# Observation of a transient intermediate in the ultrafast relaxation dynamics of the excess electron in strong-field-ionized liquid water

Pei Jiang Low[1,3], Weibin Chu[2,3], Zhaogang Nie[1], Muhammad Shafiq Bin Mohd Yusof[1], Oleg V. Prezhdo [2] ✉ & Zhi-Heng Loh [1] ✉

A unified picture of the electronic relaxation dynamics of ionized liquid water has remained elusive despite decades of study. Here, we employ sub-two-cycle visible to short-wave infrared pump-probe spectroscopy and ab initio non-adiabatic molecular dynamics simulations to reveal that the excess electron injected into the conduction band (CB) of ionized liquid water undergoes sequential relaxation to the hydrated electron $s$ ground state via an intermediate state, identified as the elusive $p$ excited state. The measured CB and $p$-electron lifetimes are $0.26 \pm 0.02$ ps and $62 \pm 10$ fs, respectively. Ab initio quantum dynamics yield similar lifetimes and furthermore reveal vibrational modes that participate in the different stages of electronic relaxation, with initial relaxation within the dense CB manifold coupled to hindered translational motions whereas subsequent $p$-to-$s$ relaxation facilitated by librational and even intramolecular bending modes of water. Finally, energetic considerations suggest that a hitherto unobserved trap state resides ~0.3-eV below the CB edge of liquid water. Our results provide a detailed atomistic picture of the electronic relaxation dynamics of ionized liquid water with unprecedented time resolution.

The interaction of high-energy radiation with aqueous solutions and biological matter leads to the ionization of liquid water. The ensuing reaction of the primary photoproducts—water radical cations and electrons— with other molecules, including the solvent, forms the basis of radiation chemistry and radiation biology[1]. The water radical cation ($H_2O^{\cdot+}$) undergoes an ultrafast proton transfer reaction to a neighboring water molecule to form the hydroxyl radical[2], a highly reactive species that triggers the oxidative damage of cellular matter[3] and corrosion in nuclear reactors[4]. The electrons, on the other hand, are postulated to induce genomic damage by dissociative electron attachment[5].

Aside from playing a central role in radiation damage, $H_2O^{\cdot+}$ and the electron also exhibit physical and chemical properties that are of fundamental interest. The electron that is injected into the solvent by ionization, for example, represents the simplest chemical species whose solvation dynamics can be investigated[6,7]. Recent studies have called into question the cavity model of the hydrated electron[8]. In addition, competing interpretations of the excited-state dynamics of the hydrated electron involving adiabatic solvation[9,10] and non-adiabatic internal conversion[11–15] have emerged. While most of these studies have centered on the nonequilibrium dynamics of the hydrated electron, prepared in the ground state prior to probing by either time-

[1]School of Chemistry, Chemical Engineering and Biotechnology, and School of Physical and Mathematical Sciences, Nanyang Technological University, Singapore 639798, Singapore. [2]Department of Chemistry, University of Southern California, Los Angeles, CA 90089, USA. [3]These authors contributed equally: Pei Jiang Low, Weibin Chu. ✉e-mail: prezhdo@usc.edu; zhiheng@ntu.edu.sg

resolved spectroscopy or nonadiabatic (NA) molecular dynamics (MD) simulations, relatively little is known about the relaxation dynamics of electrons that are directly injected into the solvent by ionization (Fig. 1).

Previous optical pump-probe experiments of ionized liquid water either attributed a lifetime of ~0.2–0.5 ps to a weakly bound electron in a shallow trap that is transiently populated en route to hydrated electron ($e_s$) formation[16–21], found no need to invoke the existence of such an intermediate[22,23], or are inconclusive[24]. The lack of consensus can be traced to several limitations associated with the previous studies. First, the ~0.3-ps laser pulses that were employed in the early studies offer lower time resolution, resulting in the large uncertainties in the formation (110–300 fs) and decay (240–545 fs) times of the intermediate state that was invoked to explain the observed dynamics[16–19]. Second, the use of relatively long laser pulses for driving ionization allow the hydrated electron that is produced by the leading edge of the laser pulse to be photoexcited by the trailing edge of the laser pulse, as pointed out in ref. 23, thus further complicating analysis. Third, the use of few-millimeter-thick sample targets housed within cuvettes in previous measurements[16–19] inevitably gives rise to cross-phase modulation artifacts that can be misinterpreted as the ultrafast response of the sample, as shown in ref. 21. Moreover, the group-velocity mismatch between the pump and probe pulses further degrades the time resolution[23]. These shortcomings have prevented a unified picture of the electronic relaxation dynamics of ionized liquid water from emerging. Nonadiabatic molecular dynamics (NAMD) simulations predict that approximately half of the conduction-band (CB) electron ($e_{CB}$) population relaxes via an electronically excited $p$-like state ($e_p$) whereas the other half relaxes directly to give $e_s$[25]. According to these simulations, the lifetime of $e_p$ is ~160 fs. Here, we report the use of few-cycle laser pulses spanning the visible to the short-wave infrared (SWIR) to elucidate the elementary electron dynamics of ionized liquid water. Our experiments strongly suggest that $e_p$ exists as a key intermediate in the ultrafast electronic relaxation process. These results are supported by NAMD simulations coupled to real-time, time-dependent density functional theory (DFT).

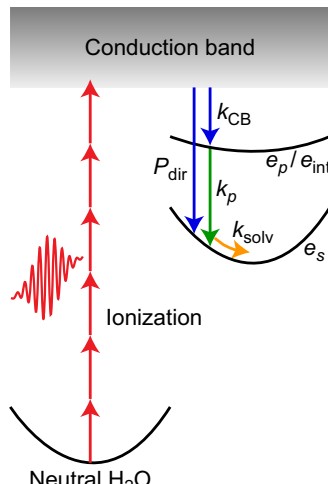

**Fig. 1 | Ultrafast electronic relaxation dynamics of ionized liquid water.** Schematic illustration of the energy levels of the excess electron in ionized liquid water, prepared by strong-field multiphoton ionization (red arrows), and its relaxation pathways. Relaxation from the conduction band (blue arrows) with rate constant $k_{CB}$ proceeds either via the $p$-like electron intermediate, $e_p/e_{int}$, or directly to the $s$-like hydrated electron ground state, $e_s$, with probability $P_{dir}$. Sequential relaxation from the $p$-like intermediate (green arrow) occurs with rate constant $k_p$. These relaxation processes yield a vibrationally hot hydrated electron, which undergoes solvation (yellow arrow) with rate constant $k_{solv}$. The involvement of the $p$-like electron intermediate in the relaxation dynamics has remained inconclusive.

## Results

In our experiments, ionization of a ~7-μm-thick liquid water microjet occurs via a strong-field multiphoton process, driven by 5-fs laser pulses with a peak intensity of $2 \times 10^{13}$ W/cm² and a center wavelength (photon energy) of 634 nm (1.96 eV) (see "Methods" for details). Pump fluence dependence measurements suggest that strong-field ionization proceeds via a four-photon resonance-enhanced multiphoton ionization process (see Supplementary Note 1 and Supplementary Fig. 1a, b). In the presence of an intense laser field, the transiently populated electronically excited states of neutral liquid water would be rapidly ionized on a sub-cycle timescale, hence limiting their contribution to the observed ultrafast dynamics. Temporal confinement of strong-field ionization to the sub-cycle timescale facilitates vertical ionization, as confirmed by previous time-resolved terahertz spectroscopy of 800-nm strong-field-ionized liquid water, revealing that an electron is injected vertically into the CB[26]. As such, it is conceivable that strong-field ionization leads to the same hydrated electron species that would be produced by radiolysis. In other words, the electronic relaxation processes that are observed in this work are also applicable to those that accompany conventional radiolysis. Differential absorption spectra were recorded with two types of probe pulses, one spanning the visible to near-infrared (NIR, 550–950 nm) and another one spanning the SWIR (1.1–1.6 μm), both of ~2-cycle duration. While strong-field ionization also leads to the ejection of electrons from the bulk liquid, these photoelectrons do not form hydrated electrons and therefore do not contribute to our transient absorption signal, which probes the electron dynamics within the bulk liquid.

The time-resolved differential absorption ($\triangle A$) spectra recorded in the visible–NIR and SWIR are shown in Fig. 2a, b, respectively. In the visible–NIR, the increase of the $\triangle A$ signal is accompanied by a blueshift of the absorption band. In the SWIR, a broad absorption band rapidly grows in after time zero, peaking at ~0.2 ps; the long-wavelength cutoff is limited by the spectral response of the InGaAs array detector used in the experiments. Experiments performed on deuterated water ($D_2O$) exhibit similar features in the time-resolved $\triangle A$ spectra. Although the $\triangle A$ time traces in the visible–NIR appear to be independent of H/D isotopic substitution (Fig. 2c), the SWIR time traces suggest slower dynamics in $D_2O$ than $H_2O$ (Fig. 2d).

The broad absorption feature that appears in the visible–NIR $\triangle A$ spectra at >1-ps time delay (Fig. 2a) is characteristic of the $e_s$ absorption, first observed in the radiolysis of water[27] and assigned to the $e_s \rightarrow e_p$ transition. Following previous work, the growth of the visible–NIR $\triangle A$ signal can be ascribed to hydrated electron formation, initially produced in the vibrationally hot state ($e_{s*}$), whose subsequent solvation leads to the blueshift of the $\triangle A$ absorption[19,23]. These studies yield time constants of ~0.2 ps for $e_{s*}$ formation and ~0.3–0.5 ps for its solvation.

The time-resolved SWIR $\triangle A$ spectra (Fig. 2b) is reminiscent of the SWIR absorption observed in previous experiments[18–20]. This feature has been attributed to a prehydrated electron residing in a shallow trap[16], also referred to as the "wet electron"[17,18,20,21] or "weakly bound electron"[19], with reported formation and decay times of 110–300 fs and 240–545 fs, respectively[16–21]. The spread of time constants suggests that a consistent interpretation of the ultrafast dynamics in the SWIR remains elusive. Moreover, the absorption maxima at ~0.85 μm[18] and ~1.3 μm[19], extracted previously from global fitting of absorption spectra, are not observed in the present work. Further compounding the difficulty in interpreting the results of previous experimental SWIR investigations is their lower time resolution (>300 fs), the possibility that the hydrated electron that is produced by the leading edge of such a long laser pulse can be photoexcited by the trailing edge of the same laser pulse, the use of ~8–9 eV for photoionization, below the ~10–11-eV energy for vertically injecting $e_{CB}$ into liquid water[28,29], and possible contribution from cross-phase modulation artifacts that arise from the use of few-mm-thick sample targets and windows[21,23]. In contrast to

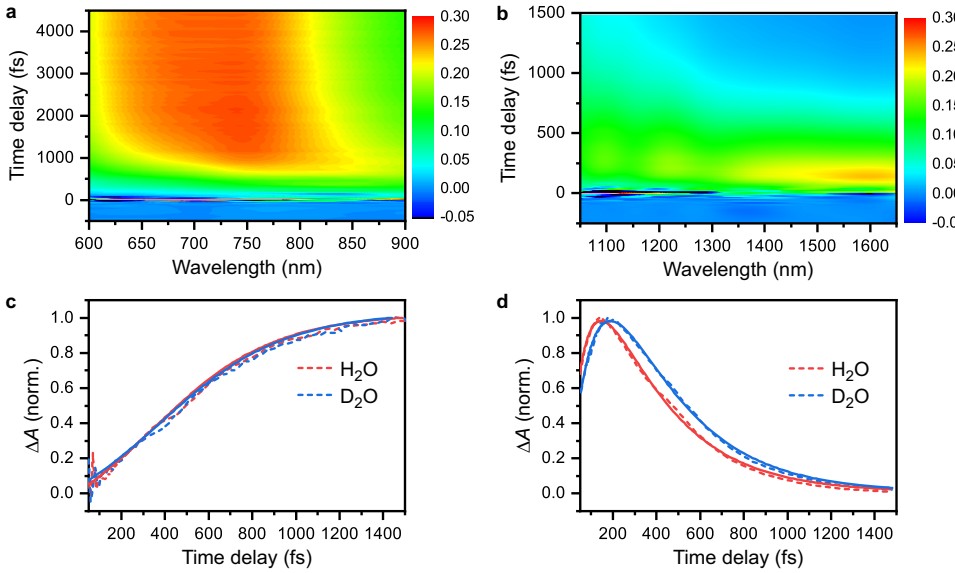

**Fig. 2 | Ultrafast optical spectroscopy of ionized liquid water.** Time-resolved differential absorption ($\triangle A$) spectra in the **a**, visible–NIR and **b** SWIR. Normalized $\triangle A$ time traces collected for $H_2O$ and $D_2O$ at **c** 700 nm and **d** 1.56 μm. The dashed lines are the experimental data and the solid lines are the fits obtained from global fitting.

these complications previously encountered in the interpretation of the SWIR kinetics, the use of few-cycle ionization and probe pulses in our experiments permit the sub-100-fs formation and subsequent decay of a pronounced SWIR absorption band to be resolved unambiguously. The transient appearance of this SWIR absorption band at early time, on top of the continuous spectral blueshift, is the experimental evidence for the existence of a transient electronic intermediate. Direct relaxation from the conduction band to the hydrated electron ground state, on the other hand, would manifest itself as a gradual appearance of the hydrated electron absorption with a time constant of ~0.3 ps, following the lifetime of the conduction-band electron, at odds with the experimentally observed prompt appearance of the SWIR absorption (see Supplementary Note 2 and Supplementary Fig. 2).

To extract the timescales that characterize the ultrafast dynamics of ionized liquid water from the experimental data, we employ global fitting to quantitatively analyze the time-resolved $\triangle A$ spectra, both in the visible–NIR and the SWIR spectral regions. Figure 1 shows a visual representation of the model used to fit the data. Aside from $e_{CB}$ and $e_s$, we consider the existence of an intermediate electronic state ($e_{int}$) whose absorption lies in the SWIR. Further details about the fitting procedure and the results are given in the Supplementary Information (see Supplementary Notes 3 and 4 and Supplementary Table 1). Global fitting yields $\tau_{CB} = 0.26 \pm 0.02$ ps and $P_{dir} \sim 0$, both independent of isotopic substitution. On the other hand, isotope-dependent behavior is observed for $\tau_{solv}$ ($H_2O$: $0.37 \pm 0.02$ ps, $D_2O$: $0.43 \pm 0.02$ ps) and $\tau_{int}$ ($H_2O$: $62 \pm 10$ fs, $D_2O$: $110 \pm 5$ fs); the pronounced isotope dependence of $\tau_{int}$ is also evident from the SWIR response (Fig. 2d).

Our value of $\tau_{CB}$ is consistent with the reported time constants for hydrated electron formation[19,22,23] and is also in good agreement with $\tau_{CB}$ ~0.2 ps directly determined from the time-resolved terahertz probing of ionized liquid water[26]. Moreover, the observed weak, but discernible H/D isotope dependence of $\tau_{solv}$ and the associated ~0.6-eV increase in the $e_s \rightarrow e_p$ transition energy due to solvation are also consistent with literature reports[19,23]. The vanishing $P_{dir}$ suggests that the entire $e_{CB}$ population is funneled through $e_{int}$ prior to forming $e_s$, in contrast to the bifurcated electronic relaxation pathways predicted by earlier NAMD simulations[25] and inferred from experiments[18,19]. In such a sequential relaxation process, $\tau_{int} < \tau_{CB}$ would imply a limited build-up of the $e_{int}$ population, hence possibly obfuscating its spectral signature

in earlier measurements with limited time resolution (see Supplementary Fig. S3).

Interestingly, the value of $\tau_{int}$ and its isotope dependence are reminiscent of hydrated electron excited-state dynamics. Time-resolved optical and photoelectron spectroscopy of $e_p$, prepared by photoexcitation of pre-equilibrated $e_s$, reveal $\tau_p$ ~50–75 fs and ~70–100 fs in $H_2O$ and $D_2O$, respectively[11–15]. The elongation of $\tau_p$ by ~1.4–1.7× in $D_2O$ relative to $H_2O$ has been attributed to the participation of librational modes in the electronic relaxation[11]. The observed $\tau_{int} \sim \tau_p$ and their similar isotope dependence strongly suggest that $e_{int}$ corresponds to $e_p$, i.e., $e_p$ exists as an intermediate state in the electronic relaxation of ionized liquid water. This is further supported by the similarity between the SWIR absorption features of ionized water and $e_p$, the latter prepared by photoexciting pre-equilibrated $e_s$ in a separate set of experiments (see Supplementary Note 5). In those three-pulse experiments, the hydrated electron $e_s$ is first prepared by strong-field-ionizing liquid water with an intense, visible–NIR laser pulse. After a time delay of 33 ps, permitting for equilibration of $e_s$, a weak, visible–NIR laser pulse is used to photoexcite $e_s$ to $e_p$, whose SWIR absorption signature is then recorded by a SWIR probe pulse. At a time delay of 40 fs, the $\triangle A$ spectrum slopes up towards the long-wavelength edge (see Supplementary Fig. 4), similar to that observed in ionized liquid water (Fig. 2b). While the $p$ state has been put forth as the transient intermediate in earlier studies, e.g., see refs. 17 and 21, we note that these assignments were made in the absence of supporting experimental evidence. On the other hand, our study resolves the lifetime of the transient intermediate as well as the isotope dependence of the lifetime, both of which are consistent with the characteristics of the $p$ state deduced by the ultrafast spectroscopy of the equilibrium hydrated electron[11–15], thus providing the first direct evidence for the assignment of the transient intermediate state to the $p$ electron.

NAMD simulations are employed to provide an atomistic model of the observed ultrafast phenomena (see "Methods" for details). We simulate the nonradiative relaxation of the photogenerated electron, starting 1-eV-deep inside the water CB, to the fully hydrated $e_s$ (Fig. 3a) via $e_p$ (Fig. 3b). Solvent rearrangement that accompanies the relaxation of the electron to the CB bottom leads to a double-cavity structure, supporting $e_p$ formation. As found in the previous studies[6,7,30–32], $e_s$ forms a single cavity, the outer boundary of which is defined by the

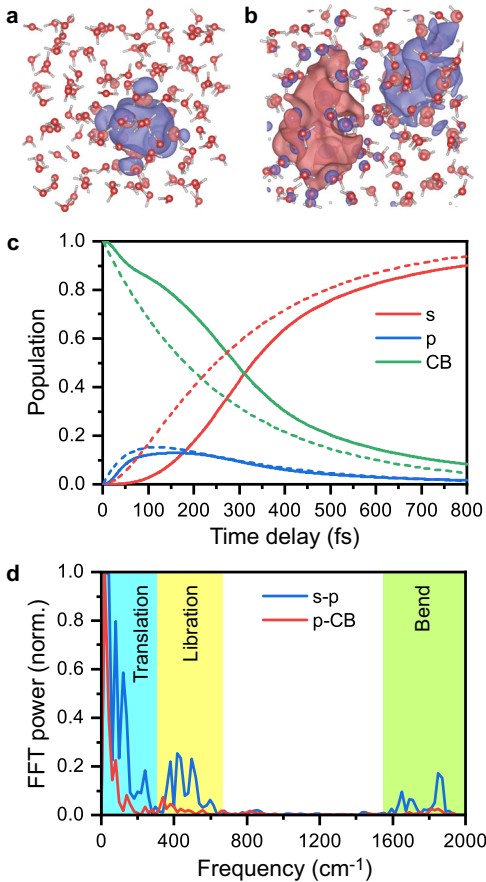

**Fig. 3 | Nonadiabatic molecular dynamics (NAMD) simulations of ionized liquid water.** Wave functions of **a**, the $s$-like ground-state hydrated electron, $e_s$, and **b**, the $p$-like intermediate, $e_p$. **c** NAMD results for the electronic relaxation from the conduction band, $e_{CB}$, to $e_p$ and then $e_s$ are shown as solid lines. Results obtained from sequential kinetics, calculated based on the experimentally determined rate constants, are shown as dashed lines. **d** Fourier transforms of the energy gaps between $e_s$ and $e_p$, and $e_p$ and $e_{CB}$, characterizing the vibrational motions responsible for the electronic relaxation.

first solvation shell. Earlier work used either single-particle pseudo-potentials to study NAMD of the photoexcited hydrated electron[6,9,25] or ab initio methods to characterize the adiabatic solvation dynamics and equilibrium structure of the hydrated electron, hence neglecting the nonadiabatic dynamics[30–32]. The current work employs NAMD coupled with ab initio time-domain DFT description of the electron dynamics. The time traces of the $e_{CB}$, $e_p$, and $e_s$ populations obtained from NAMD (solid lines in Fig. 3c) match well the solutions of the sequential kinetics $e_{CB} \rightarrow e_p \rightarrow e_s$, evaluated based on the experimentally determined time constants (Fig. 3c, dashed lines).

Fourier transforms of the energy gaps between $e_{CB}$, $e_p$, and $e_s$ (Fig. 3d) characterize the phonon modes responsible for the non-radiative relaxation of the photogenerated electron. Relaxation from $e_{CB}$ to $e_p$ is governed exclusively by the low-frequency hindered translational motions of water (Fig. 3d, red line), consistent with the expectation that electronic transitions between these energetically closely spaced states require the transfer of small amounts of electronic energy to the vibrational subsystem during each individual transition, a process well-accommodated by slow hindered translational modes. Moreover, translational motions of water away from the electron that is being hydrated[9] promote cavity formation to accommodate $e_p$, which in turn accounts for the observed similarity in the characteristics of $e_{int}$ and $e_p$. In comparison, nonradiative $p$-to-$s$ relaxation exhibits contributions from librational (hindered

rotational) motions of water molecules and even high-frequency intramolecular bending (Fig. 3d, blue line), consistent with the experimentally observed H/D isotope effect. Contributions of the low-frequency hindered translation remain strong as well. The hindered translational and librational motions facilitate the shift from a double-lobe cavity shape for $e_p$ (Fig. 3b) to the single lobe for $e_s$ (Fig. 3a)[11]. Participation of the higher frequency modes assists in accommodating a larger amount of energy dissipated during the $p$-to-$s$ transition across an energy gap[25].

The ab initio NAMD simulations demonstrate that the $p$ state contains two lobes, as expected, and that the two lobes occupy two adjacent cavities (Fig. 3b). The earlier NAMD simulations based on a single-particle pseudopotential description of the hydrated electron showed that the $p$ state occupied an elongated cavity that turned into a spherical cavity upon relaxation to the $s$ state[9]. The current ab initio DFT level of theory, which includes not only the excess electron but also the valence electrons of all the water molecules, produces a somewhat different structure. The two lobes of the $p$ state are separated by a low electron density region that allows a few water molecules to penetrate between the lobes. This double-cavity-type structure occurs because the $p$ state has a node in the middle, and in the absence of electron density at the node, there is no Pauli repulsion that pushes electrons of water molecules away from this region. The pseudopotential approach employed in the earlier studies either creates an elongated single cavity[9] or no cavity[8], depending on the pseudopotential parameters. The more sophisticated ab initio description that includes all the valence electrons allows the double cavity to be formed, with water molecules pushed away from the regions of high $p$-electron density and allowing water molecules to penetrate regions of low $p$-electron density near the node.

## Discussion

In the following, we consider the extent to which our experimental data is consistent with alternative electronic relaxation mechanisms that have been proposed. First is the direct relaxation[22,23] of $e_{CB}$ to $e_{s^*}$. The corresponding simulated spectrally resolved transient absorption (see Supplementary Fig. 2) is missing the prominent SWIR absorption band that is clearly observed in the experiments (Fig. 2a). The appearance of the SWIR transient absorption at early time delays, beyond the simultaneous spectral shift and the growth of the $e_s$ absorption, necessitates the inclusion of a transient intermediate in the kinetic model. Second, instead of the $p$ state, we consider the existence of the hypervalent hydronium radical species, $H_3O\cdot$, as an intermediate state. In previous time-resolved photoelectron spectroscopy (TRPES) studies, the formation of $e_{s^*}$ following either 9.3-eV photoexcitation of water clusters, $(H_2O)_N$ ($\langle N \rangle \sim 400$)[33], or 7.7-eV photoexcitation of liquid water[34] was attributed to the photodissociation of $H_2O$ to yield the hypervalent $H_3O\cdot$ radical with Rydberg-like character, followed by subsequent autoionization, i.e.,

$$H_2O^*(aq) + H_2O\,(1) \rightarrow H_3O\cdot(aq) + OH\cdot(aq) \quad (1)$$

$$H_3O\cdot(aq) \rightarrow H_3O^+(aq) + e_{s^*}(aq) \quad (2)$$

This mechanism yields $H_3O^+$ and $e_{s^*}$ without passing through the $e_p$ transient intermediate. For water clusters, timescales for $e_{s^*}$ formation via this channel were found to be 43 fs and 61 fs for $H_2O$ and $D_2O$ clusters, respectively[33]. In our experiments, resonance-enhanced strong-field ionization via the neutral $\widetilde{A}$ state could initiate these dynamics. The subsequent formation of $H_3O\cdot$ as a transient intermediate is consistent with the kinetic scheme inferred from our experiments. However, we note that ab initio calculations predict a strong optical absorption signature[35] at ~780 nm for hydrated $H_3O\cdot$, inconsistent with the experimentally observed absorption band at

~1.6 µm. Moreover, the formation of $H_3O·$ by the photodissociation of the water O−H bond is expected to occur on the sub-10-fs timescale if we consider the repulsive nature of the $\tilde{A}$ state accessed by ~8-eV photoexcitation[36,37]. In contrast, the intermediate state observed in our experiment forms on longer timescales (62 fs in $H_2O$ and 110 fs in $D_2O$). For these reasons, we exclude the possibility of identifying $H_3O·$ as the transient intermediate. Third, instead of producing the CB electron via vertical ionization, we consider the possibility that the $\tilde{A}$ state autoionizes to yield $e_{CB}$[38], notwithstanding the fact that autoionization needs to be ultrafast to compete with the dissociation of the $\tilde{A}$ state. The CB electron produced by autoionization then undergoes sequential relaxation process to $e_s$ via $e_p$. Our experiments are unable to distinguish the formation of $e_{CB}$ via the autoionization or vertical ionization channels because they do not probe the spectroscopic observable of the CB electrons, located in the terahertz[26]. Future experiments that probe the terahertz absorption of ionized liquid water can elucidate the existence of the autoionization channel via the delayed appearance of $e_{CB}$.

It is interesting to compare our experimental results to those obtained from recent TRPES studies of ionized water clusters[33,39,40] and liquid water[34]. Water clusters, $(H_2O)_N$ ($\langle N \rangle \sim 400$), that were prepared in ref. 33 were ionized with either 9.3- or 15.5-eV pulses whereas refs. 39 and 40 reported the 7.7- or 10.9-eV ionization of $(H_2O)_N$ ($\langle N \rangle \sim 300$). In contrast to the experiments on water clusters, ref. 34 reported the ionization of a liquid microjet by either 7.7-, 9.3-, or 11.0-eV pulses. Approximating water as a wide-band-gap semiconductor[41–43], with an estimated vertical (direct) bandgap[28,29] of ~10–11 eV and an adiabatic (indirect) bandgap[44,45] of ~7 eV, these photoexcitation energies favor either vertical ionization (10.9 eV, 15.5 eV) or adiabatic ionization (7.7 eV, 9.3 eV) of water. TRPES reveals sub-picosecond to picosecond evolution of the electron binding energies and linewidths, attributed to electron hydration dynamics and found to be independent of excitation energy[33,34,39,40]. In addition, ref. 33 resolves a sub-50-fs delay in the peak of the $e_{s*}$ photoelectron signal, ascribed to $H^+$ and H atom transfer upon vertical and adiabatic ionization, respectively. On longer, ~10-ps timescales, the decay of the TRPES signal following the depopulation of hydrated electrons by geminate recombination is observed. These results suggest that the electron produced by ionization relaxes directly to the $s$ ground state of the hydrated electron, albeit in a vibrationally excited $s^*$ state, without passing through the intermediate $p$ state. Possible explanations for the discrepancy between the TRPES results and ours are discussed below.

First, we consider the experiments performed on clusters $(H_2O)_N$, which have estimated radii of 1.3 nm ($\langle N \rangle \sim 300$)[39,40] and 1.4 nm ($\langle N \rangle \sim 400$)[33]. Irradiation with 10.9-eV and 15.5-eV light yields ejection lengths of $\langle r_0 \rangle \sim$ 3.2 nm and ~3.8 nm, respectively[38], both of which exceed the radii of the water clusters. As such, it is conceivable that the hydrated electron and its precursor, if any, are initially localized on the surface of the cluster. The abundance of surface defects and the low solvent reorganization energy requirements further favor the formation of a hydrated electron at the surface[46]. Considering that a surface-bound, ground-state hydrated electron has a low binding energy of ~1.6 eV[46] or less[47], and that its internalization and concomitant increase in binding energy occurs on a timescale of ~0.5 ps[47], the $p$ excited state of the hydrated electron might not even exist as a bound state on the surface if we assume that the $s$–$p$ energy gap on the surface is unchanged from the bulk value of 1.7 eV. Under such circumstances, one would expect the $p$ state to be energetically inaccessible during the electronic relaxation of ionized water clusters, i.e., the CB electron relaxes directly to the $s^*$ state. Second, our use of strong-field ionization favors the injection of an electron vertically into the conduction band. In contrast, the 7.7- and 9.3-eV photoexcitation employed in refs. 33,40 and 34 lie below the threshold required to inject an electron vertically into the CB. In those cases, the observed ultrafast dynamics is explained in terms of $H_2O$ undergoing sub-50-fs photodissociation to

yield the $H_3O·$ radical[33,34] (Eq. (1)), which then autoionizes to yield $H_3O^+$ and $e_{s*}$ (Eq. (2)), thus circumventing the $e_p$ intermediate entirely. Third, the limited time resolution of the experimental setups (~70–220 fs FWHM)[33,39,40] and possible coherent artifacts complicating the analysis of the early-time dynamics[34] might have obfuscated signatures of the short-lived $p$ state in the time-resolved photoelectron spectra.

That the excited $p$ state of the hydrated electron exists as an intermediate state in the electronic relaxation dynamics of ionized liquid water appears counterintuitive for the following reasons. First, the dynamics of ionized liquid water and those of the photoexcited hydrated electron involve different initial solvent configurations. In the case of ionized liquid water, the initial solvent configuration is that of liquid water at equilibrium, whereas in the case of the excited-state dynamics of the hydrated electron, the hydrated electron initially resides in a solvent cavity of dimensions comparable to the radius of gyration of the hydrated electron (2.44 Å)[7]. Hence, our assignment of the intermediate state to the $p$ state suggests that solvent reorganization within the timescale for relaxation from the CB (~0.3 ps) yields the solvent configuration of the hydrated electron. This is supported by the ab initio NAMD simulations, which show relaxation of the high-energy excited electron from deep inside the CB to the $p$ shaped state exhibiting two lobes (Fig. 3b). Second, ionized liquid water possesses the highly reactive $H_2O^{·+}$ species[2,48–50] in addition to the injected electron. While our present study focuses on electronic relaxation dynamics, it is important to realize that the valence hole created by ionization undergoes ultrafast temporal evolution at the same time. According to ab initio molecular dynamics simulations, the initially delocalized valence hole localizes on the ~30-fs timescale onto a single water molecule, forming the $H_2O^{·+}$ radical cation[2,50], which subsequently undergoes proton transfer to a neighboring water molecule to produce the hydronium ion, $H_3O^+$, and the hydroxyl radical, OH·. A recent femtosecond soft X-ray absorption study found a timescale of ~50 fs for this ultrafast proton transfer reaction[2]. An obvious question that arises is the extent to which electronic relaxation dynamics is affected by hole localization and subsequent proton transfer. The latter is accompanied by the contraction of the intermolecular O···O distance between $H_2O^{·+}$ and the neighboring $H_2O$ molecules in its immediate vicinity[2]. The contraction of the O···O bond (to 2.4 Å) promotes proton transfer, whereupon the O···O distance returns to that of equilibrium neutral water (2.7 Å). It is conceivable that the valence hole-induced solvent reorganization dynamics could interfere with the electronic relaxation dynamics. However, we note that our ab initio NAMD simulations reproduce the timescales for the electronic relaxation dynamics even though they do not consider the competing hole dynamics. Moreover, an ejection length of ~40 Å, determined from a recent time-resolved terahertz study of strong-field-ionized liquid water[26], implies that the initial ionization site and the site at which the electron localizes to form the $p$ state is separated by a distance of ~16 solvent shells. The large electron-hole separation produced by strong-field ionization provides a plausible explanation for the absence of any interference between the electron and hole dynamics.

Beyond the ultrafast dynamics, our experimental observation of the SWIR absorption of $e_p$, with transition energy $E_p$ ($H_2O$: $0.754 \pm 0.031$ eV, $D_2O$: $0.752 \pm 0.023$ eV), can be combined with the $e_s \rightarrow e_p$ transition energy, $E_s(\infty)$, and the literature values for the vertical electron affinity, $V_0$, and vertical detachment energy, VDE, to construct an energy level diagram for the excess electron in ionized liquid water (Fig. 4). In the diagram, AEA is the adiabatic electron affinity, or the enthalpy of hydration of the electron[51], and $Q_{solv}$ is the solvent coordinate for cavity formation to accommodate $e_s$. $V_0$ represents the energy difference between the CB minimum and the vacuum level of liquid water at its equilibrium solvent configuration, whereas VDE is the energy that is needed to detach a ground-state hydrated electron without

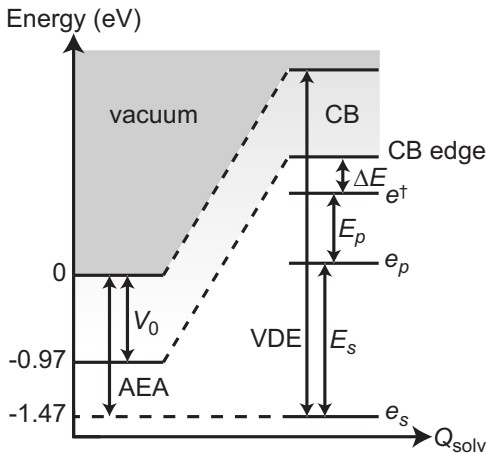

**Fig. 4 | Electronic energy level diagram for the excess electron of ionized liquid water.** The optical transition energies for $e_s$ and $e_p$ that are obtained from the present work, denoted $E_s$ and $E_p$, respectively, are combined with other energetic parameters for the excess electron in water to reconstruct this energy level diagram. In the diagram, $Q_{solv}$ is a generalized solvation coordinate, AEA and $V_0$ are the adiabatic and vertical electron affinities of liquid water, respectively, and VDE is the vertical detachment energy of the hydrated electron. The result points to the existence of an energy level, $e^\dagger$, hitherto unobserved in experiments, that lies $\triangle E$ - 0.3 eV below the conduction-band (CB) edge.

solvent reorganization. While a critical discussion of the literature values for $V_0$ and VDE lies beyond the scope of this work, we note that a set of consistent VDE values has emerged only recently with the introduction of corrections for energy-dependent electron scattering within the liquid or cluster[29,39,52,53]. These studies yield VDE values of $3.70 \pm 0.10$ eV[52], $3.70 \pm 0.15$ eV[39], $3.76 \pm 0.05$ eV[53] and $3.77 \pm 0.10$ eV[53]; without correcting for electron scattering, the retrieved VDEs are generally lower, in the range of 3.2–3.7 eV[54–57]. On the other hand, consensus for the value of $V_0$ has remained elusive. An early estimate gave $V_0 = 0.75$ eV, whereas a recent study to determine the hydrated electron VDE[52] employed $V_0 = 1.0$ eV in its data analysis[58]. First-principles calculations performed with different-sized simulation cells and at different levels of theory have yielded $V_0$ values of 0.2 eV[43], 0.5 eV[59], 0.97 eV[30], and 1.1 eV[60]. The difficulty in obtaining an accurate value of $V_0$ can be seen from the fact that path-integral molecular dynamics simulations combined with GW calculations yield both the smallest (0.2 eV)[43] and largest (1.1 eV)[60] values of $V_0$: non-self-consistent $G_0W_0$ calculations give $V_0 = 0.2$ eV whereas self-consistent GW calculations with implicit vertex corrections, albeit on a smaller simulation cell (32 vs. 64 water molecules) than in ref. 43, give $V_0 = 1.1$ eV. We note that the calculated $V_0$ value of 0.2 eV is at odds with the results of hydrated electron de-trapping experiments, which reveal that one-photon, 3.1-eV photoexcitation of the hydrated electron accesses the CB[61]. In view of this discrepancy, we employ $V_0 = 0.97$ eV, obtained from hybrid density functional theory combined with either thermodynamic integration[30]. Assuming that $V_0$ is independent of $Q_{solv}$[38], we infer that the final state that is accessed by the SWIR probe pulse, $e^\dagger$, resides $\triangle E = VDE - V_0 - E_p - E_s(\infty)$ below the bottom of the CB. With VDE = $3.76 \pm 0.05$ eV[53], $V_0 = 0.97$ eV[30], and the experimentally determined $E_p = 0.754 \pm 0.031$ eV and $E_s(\infty) = 1.730 \pm 0.019$ eV, we obtain $\triangle E = 0.31 \pm 0.06$ eV. (Note that $\triangle E$ represents a lower bound because the low-energy cutoff in the spectral response of the SWIR detector imposes an upper bound on the retrieved $E_p$.) The nature of $e^\dagger$ is unknown, as previous studies that report a SWIR absorption signature[16–21] do not address the identity of the final state. It is unclear if $e^\dagger$ is an intermediate in the electronic

relaxation of ionized liquid water. Our global fitting does not consider the existence of $e^\dagger$ since its signature is not observed within our probe spectral range. One possible candidate for $e^\dagger$ is a shallow trap state, recently uncovered in ab initio MD simulations to reside 0.26-eV below the CB of liquid water[31], in apparent agreement with the range of $\triangle E$ deduced herein. According to the simulations, however, the existence of this trap state precedes cavity formation[31]; as such, $e^\dagger$ is not necessarily accessible from the equilibrium solvent configuration of $e_p$ or $e_s$. It will be interesting for future studies to address the nature of $e^\dagger$.

Our results identify $e_p$ as a transient intermediate in the electronic relaxation dynamics of ionized liquid water. In addition, energetic considerations point to the existence of a shallow trap state located ~0.3-eV below the CB edge. The presence of this state could be established in future experiments by employing single-cycle mid-IR pulses[62] to observe its absorption signature. Another possibility is to harness femtosecond soft X-ray absorption spectroscopy performed at the oxygen K edge. Indeed, a recent study that focused on the hole dynamics of ionized liquid water uncovered manifestations of electron dynamics in the pre-edge absorption region[2], which evolves with a time constant of $0.26 \pm 0.03$ ps, in good agreement with $\tau_{CB}$ deduced from the present study. Combining the data reported herein with the ultrafast dynamics observed in future mid-IR or X-ray absorption spectroscopy experiments will furnish a more complete picture of the electronic relaxation dynamics of ionized liquid water.

## Methods
### Experimental
The experimental setup for transient absorption spectroscopy is based on a chirped-pulse-amplified Ti:sapphire laser system (Coherent, Legend Elite Duo-USX) that outputs 4.5-mJ, 30-fs pulses at 1-kHz repetition rate and 800-nm center wavelength. The laser pulses are directed to a helium-filled hollow-core fiber, where the pulses undergo spectral broadening by self-phase modulation. The pulses are then temporally compressed by a set of chirped mirrors (Ultrafast Innovations GmbH, PC1332). Following chirped mirror compression, 80% of the output, with ~0.6 mJ pulse energy, is directed into a visible–NIR $4f$ pulse shaper (Biophotonics Solutions, FemtoJock-P) that is equipped with a 128-pixel spatial light modulator for adaptive pulse compression before being directed to the sample. Experiments in the visible–NIR employ broadband pump and probe pulses, with a wavelength range that spans 528–919 nm at −20 dB and a carrier wavelength of 757 nm. Interferometric autocorrelation measurements, performed at the position of the sample, reveal a pulse duration of 4.5 fs (see Supplementary Note 6 and Supplementary Fig. 5a, b); pronounced modulations in the visible–NIR spectrum, however, limited the compressibility of the laser pulses, thus giving rise to pedestals that extend to ~70 fs in the inteferometric autocorrelation trace. In the case of the experiments that employ SWIR probing, the spectrum of the visible–NIR pump pulse is truncated to 533–741 nm by blocking the long-wavelength components of the hollow-core fiber output at the Fourier plane of the pulse shaper. The resultant output pulses have a carrier wavelength of 634 nm and are compressed to 5.4 fs (see Supplementary Note 6 and Supplementary Fig. 5c, d). Short-wave infrared (SWIR) probe pulses are produced by difference frequency mixing of the short- and long-wavelength components of the compressed hollow-core fiber output in a 2-mm-thick, Type II BBO crystal cut at $\theta = 36.7°$ (Castech). After transmission through a 1000-nm long-pass Schott glass filter (Thorlabs, FGL1000) and an ultrabroadband wire-grid polarizer (Thorlabs, WP12L-UB), the SWIR is directed to another $4f$ pulse shaper (Biophotonics Solutions, FemtoJock-P) that is equipped with a 320-pixel spatial light modulator for adaptive pulse compression. The spectral range of the SWIR probe pulse spans 0.98–1.70 μm, with a carrier wavelength of 1.37 μm, and is compressed to 8.5 fs (see Supplementary Note 6 and Supplementary Fig. 5e, f). In all cases, the

pulses that are employed in the experiments are ~2–3 cycles in duration.

For experiments that probe in the visible–NIR (SWIR), the pump pulses have pulse energies of 38 μJ (21 μJ) and are focused by a 50-cm focal length spherical mirror to a beam waist ($1/e^2$ radius) of 84 μm (113 μm) at the sample target. On the other hand, the visible–NIR (SWIR) probe pulses have pulse energies of 22 nJ (8 nJ) and are focused by a 15-cm focal length off-axis parabolic mirror to a beam waist ($1/e^2$ radius) of 26 μm (24 μm). The focal spot sizes of the pump beams are ~3–4× larger than those of the probe beams, hence minimizing the effect of spatial averaging on the observed ultrafast dynamics. An optical delay stage line that is driven by a piezo translation stage (Physik Instrumente, N-664.3 A) is positioned in the pump beam path to provide a variable time delay between the pump and probe pulses. In the vicinity of time zero, the time delay is varied in steps of 2 fs between −100 fs to +200 fs; beyond this range, a step size of 10 fs is employed. An optical chopper that is synchronized to the 1-kHz pulse train of the laser is positioned in the pump beam path before the sample to modulate the repetition rate of the pump beam at 0.5 kHz.

The sample target comprises a microjet that is produced by a slit-type nozzle (Metaheuristics, Type L). Experiments on liquid water employ distilled water, whereas experiments on deuterated water employ 99.8% $D_2O$ (Cambridge Isotope Laboratories) that were used as received. A peristaltic pump (Cole-Parmer, 07528-10) is used alongside a pulse dampener and the slit nozzle to produce a liquid microjet with a stable volume flow rate of 92 mL min$^{-1}$, corresponding to a vertical flow rate of ~8.5 mm ms$^{-1}$. The vertical flow rate ensures that a fresh volume of liquid is exposed to each ionization pump pulse, therefore preventing the growth of spurious signals due to the accumulation of photoproducts in the focal volume. By employing spectral interferometry, the liquid microjet path length is found to be 7 μm, sufficient for producing a sizeable differential absorption signal while minimizing the temporal broadening of the laser pulses in the jet medium due to dispersion. In addition, the thin liquid jet limits the degradation of time resolution due to group-velocity mismatch between the visible pump and SWIR probe pulses.

In the case of the visible–NIR probe experiments, a reflective neutral density filter (Newport, FRQ-ND05) is used as a beamsplitter to produce the probe and reference beams. The probe beam that transmits through the microjet sample target is focused into a 300-mm spectrograph (Princeton Instruments, Acton SP2300), equipped with a 150-grooves per mm grating blazed at 800 nm. The spectrograph is equipped with a 1024-pixel silicon photodiode array detector (Stresing, Hamamatsu S8381-1024Q) with a read-out rate of 1 kHz, which allows simultaneous acquisition of the probe spectra obtained with and without the ionizing pump pulse. Reference probe pulses are focused into a second, identical spectrograph that is equipped with the same gratings and photodiode array detector. The use of single-shot referencing reduces noise caused by shot-to-shot fluctuations of the probe pulse. Experiments in the SWIR do not employ single-shot referencing. The SWIR probe beam that is transmitted through the sample is focused into a 300-mm spectrograph (Horiba Jobin-Yvon, iHR320), equipped with a 150-grooves per mm grating blazed at 1.2 μm. The spectrograph is equipped with a 512-pixel InGaAs photodiode array detector (Stresing, Sensors Unlimited SU512LDB-1.7T2-0500), Peltier-cooled to −20 °C, with a read-out rate of 1 kHz.

Experiments were performed by recording the differential absorption signals with both parallel ($\triangle A_{\parallel}$) and perpendicular ($\triangle A_{\perp}$) relative polarizations between pump and probe beams. From these measurements, the isotropic differential absorption signal is reconstructed via the relation $\triangle A_{iso} = \left(\triangle A_{\parallel} + 2\triangle A_{\perp}\right)/3$. To focus on the population dynamics, all our subsequent analyses are performed using $\triangle A_{iso}$. Differential absorption spectra are processed by applying a Butterworth long-pass filter (see Supplementary Note 7) to remove spurious features caused by the presence of modulations in the probe

spectra. The relatively wide absorption spectra of the hydrated electron justify this post-processing method.

## Theoretical

The ab initio MD simulations are carried out with PBE0 hybrid functional[63] using the auxiliary density matrix methods (ADMM)[64] as implemented in CP2K[65]. The bulk water is represented with 128 $H_2O$ molecules at the experimental density. To simulate the dynamics of the hydrated electron, the MD simulations are performed with an extra electron through spin-polarized calculation. A 0.5 fs time step is used for all simulations. The core–valence interactions are described through Goedecker–Teter–Hutter (GTH) pseudopotentials, and the valence electron wave functions are expanded in a triple-ζ valence polarized (TZV2P) basis. The fraction α of the Hartree–Fock exchange is increased to 0.40 to achieve a good description of the bandgap and the dynamics, as done in the hydrated electron studies[30,31]. The van der Waals interactions are considered with the rVV10 functional[66].

NAMD simulations are performed to model relaxation of the photoexcited electron from the initial state down the CB to the p and then s states using the PYXAID software[67,68]. The neutral bulk water is equilibrated at room temperature for 5 ps, corresponding to the initial conditions of the experiments. Then, an electron is added 1 eV above the lowest unoccupied Kohn–Sham orbital of neutral bulk water, representing the initial photoexcitation, and NAMD simulations are carried out using the fewest switches surface hopping method[69,70].

To characterize the phonon modes responsible for the relaxation, we examine the time-resolved vibrational spectra, or more specifically, the electron-phonon influence spectra. These spectra are obtained from Fourier transforms of the energy gaps between the initial and final states, i.e., between the lowest energy conduction-band state and p state, and between the p and s states. To obtain time resolution, the Fourier transforms are computed for the parts of the trajectories in which the electron is already relaxed to the initial state for each transition. Because these trajectory parts are relatively short, the spectral resolution is relatively low. Nevertheless, the influence spectra can clearly identify what kind of modes—translational, librational, internal bending—are participating in a particular transition because the frequencies of these motions differ by hundreds of cm$^{-1}$.

## Data availability

The data generated in this study have been deposited in the DR-NTU database and are available at https://doi.org/10.21979/N9/FVLTAJ.

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

## Acknowledgements

We acknowledge financial support from the Ministry of Education, Singapore (grants MOE-T2EP50221-0004, RG1/20, RG105/17, MOE2014-T2-2-052 to P.J.L, Z.N., M.S.B.M.Y., and Z.-H.L.), Nanyang Technological University (Nanyang President's Graduate Scholarship to M.S.B.M.Y.), and the U.S. National Science Foundation (grant CHE-2154367 to O.V.P.).

## Author contributions

P.J.L., Z.N., and M.S.B.M.Y. developed the experimental apparatus, P.J.L. performed the experiments and analyzed the data, W.C. performed the theoretical simulations, Z.-H.L. conceptualized the project, Z.-H.L. and O.V.P. supervised the project, Z.-H.L., O.V.P., P.J.L., and W.C. wrote the manuscript with inputs from all authors.

## Competing interests

The authors declare no competing interests.
