## [Peer Review File · Nature Communications]

Reviewers' comments:

Reviewer #1 (Remarks to the Author):

I'm sorry to say that I'm not excited about that paper. It deals with an old problem, i.e. the relaxation of the aqueous electron after photo-ionization of water. The experimental observation of a transient species in the NIR per se is not new, as the authors admit. It probably started with Ref 23 from 1997, where similar observations have been made. When I understand the authors correctly, what is supposed to be new is the interpretation of that transient species. They claim that it is a p-like wavefunction, rather than a continuously solvating s-like state. I however don't see how their new results would support that conclusion, i.e. in what sense a continuously moving absorption band contradicts their observation, while a discrete intermediate is in agreement. Independent of that, I'm pretty sure that p-like wavefunctions have been put forward before to explain this observation.

To further support their conclusion, the present "NAMD simulations". First, the level of theory is in essence not described in the main paper, but that would be very crucial as it has been shown in the past that the properties of the solvated electron very strongly depend on the level of description. Second, I don't understand how the assignment is done. The authors refer to Fig.3b, which shows a red and a blue lobe, which are called charge density in the figure caption. The two lobes of a p-orbital have opposite sign, but that is NOT charge density. I must misunderstand something completely.

In conclusion, the paper needs quite some attention to work out better the arguments. Independent of that, after 25 years of work on this question, I would no longer consider it the most exciting science.

Reviewer #2 (Remarks to the Author):

The manuscript entitled "Observation of a transient intermediate in the ultrafast relaxation dynamics of the excess electron in ionized liquid water" by Low et al. (performed at the laboratory of prof. Loh and accompanied by theory calculations by Oleg Prezhdo) focuses on the electron formation upon multiphoton ionization or maybe excitation of liquid water. Even sixty years after its observation during pulse radiolysis, the hydrated electron remains to be somewhat enigmatic. For example, in 2010, the group of Benjamin Schwartz provocatively suggested a radically different view

of the hydrated electron structure. The Science paper brought a huge excitement, with many research groups demonstrating that the outcome was just an artifact of poor electronic structure description. I make this lengthy historical introduction to emphasize that the general attractiveness of the subject should not compromise the attempt to understand in detail all the aspects of the theory and experiment. Below, I summarize some aspects I did not understand – and I might be easily wrong. Yet I would be happy to read the author's reaction.

My first question is whether the authors are sure what states are formed initially during the radiolysis. The strong field used in the work can electronically excite the molecule as well as ionize it. The excitation seems to lead to the observation of the very same hydrated electron. The authors should discuss in detail this aspect.

Relatively recently, the same process was investigated by time-resolved photoelectron spectroscopy in the laboratory of Hans-Jacob Wörner at ETH (see <https://www.science.org/doi/10.1126/sciadv.aaz0385>). TRPES is generally assumed to provide the most direct insight into the electronic structure, with both the energetics and anisotropy revealing the details of the dynamics. The initially formed electronic states seem to be much better selected in this case, demonstrating that both the excitation and ionization lead to the same product on a similar time scale. The interpretation is, naturally, different for both processes. The authors do not seem even aware of this work yet it seems to be an important anchor to their new study on a multiphoton generation of excess electrons. The work might be not fully relevant as it is done on clusters that are arguably solid-like. But is the difference large enough to ignore the comparison?

It would be fair to place the strong field generation of the hydrated electron in the title – it is one of the aspects marking a distinction from previous papers.

The experiments are accompanied by ab initio MD calculations. I found this part confusing. The theory is not described in the main text, the approach is not justified. The way how the time-resolved vibrational spectra were obtained needs for sure a deeper discussion. I think that Nature Communications is not particularly restrictive on the size of the manuscript and all details needed to reconstruct the simulations should be provided.

Some minor issues. The authors cite Sanche's hypothesis on the role of pre-solvated electrons in DNA damage. It should be fair to mention that the community remains unconvinced (and I like the hypothesis).

Figure 4 should be augmented, adding especially all recent photoemission data.

To conclude, I find the study interesting yet more work should be invested into placing the present finding in a context. The study could be also more “honest” in the sense that alternative hypotheses to explain the data are used.

Reviewer #3 (Remarks to the Author):

This is certainly a nice work providing some new insight into the ultrafast relaxation dynamics of the excess electron in ionized liquid water. Specifically, the authors suggest that the entire population of electrons injected into the water conduction band goes through a previously (unconfirmed) trap state in the band gap to subsequently form the equilibrated hydrated electron. I consider the new spectroscopic data, largely due to improved temporal resolution, and the accompanying theoretical computations, an important contribution in advancing our understanding of the respective ultrafast electronic relaxation processes, including their mechanistic details.

However, the results need to be discussed in much greater depth which includes to thoroughly relate to a number of findings from previous works, not addressed in the present manuscript. With the extended discussion the manuscript won't fit format the communication format, also considering the many necessary assumptions (I come back to that) the authors need to make in order to reach at their interpretation of the results. As much as I appreciate the authors' overall work, there is room for alternative conclusions. Furthermore, I am quite disappointed how little the authors comment on water being a large-band-gap semiconductor and the implications in the present context. There are also several experimental aspects that need to be addressed more thoroughly. In the following I provide some specific comments.

Pages 3,4: Regarding the multi-(1.96 eV) photon ionization, I wonder how well the order of the process can be controlled. My concern is that 'ionization' is not limited to the injection of electrons deep into the CB but also produces true photoelectrons, emitted into vacuum. This additional channel would presumably affect the interpretation of the data, and should be commented on. A related issue is if strong-field ionization can be considered to be equivalent to single-photon ionization with regard to the relaxation dynamics of the excess electron. To my understanding this is not obvious at all, and there may be a possibility that this has a signature in the experimental data.

Page 5: The authors say: “Moreover, the absorption maxima at $\sim 1.3 \mu\text{m}$ and $\sim 0.85 \mu\text{m}$, extracted previously from global fitting of absorption spectra, are not observed in the present work.” Is that a

correct statement? There seems to be intensity near 1.3 μm , and the 0.85 μm range is not covered in Figure 2. The authors should explain the reason for this discrepancy (if real).

On the same page it also says: "...the use of $\sim 8 - 9$ eV for photoionization, below the ~ 10 -eV energy for vertically injecting eCB into liquid water^{26,27} ..." This is the point where I wished the authors attempted to consider liquid water as a semiconductor, and quantitatively relate to the size of water's band gap. This should be inspired by works of the Sprik (C. Adriaanse et al., JPCL 3, 3411, 2012) and Galli (A. Gaiduk et al., Nat Comm 9, 247, 2018) groups. Furthermore, as I had pointed out above, 10 eV photon energy corresponds also to the onset of producing true photoelectrons. How are these processes related, and how do they potentially affect each other? Wouldn't it be useful to explicitly address and distinguish these two channels? In fact, even the title of this manuscript can be misunderstood.

Page 7: It says: "The observed $\tau_{\text{int}} \sim \tau_p$ and their similar isotope dependence strongly suggest that e_{int} corresponds to e_p , i.e., e_p exists as an intermediate state in the electronic relaxation of ionized liquid water." I find this a very important result which should be discussed in greater detail though, expanding on the role of H_2O^+ , and certainly that text should not be moved to the SI but rather be part of the Discussion section. Furthermore, I believe that the concept of a double-cavity structure is new, and the authors should emphasize this aspect further.

Pages 9,10: I am quite puzzled by the energy values that were chosen on page 9 (bottom), and then used to construct Figure 4. Specifically, why have the authors chosen $\text{VDE} = 3.76$ eV and not lower values? But the bigger worry is the $\text{V}_0 = 0.97$ eV value. Admittedly, this value has been debated for a long time. Not mentioning that it is likely much smaller (Gaiduk et al.) is not helpful since there are alternatives to construct such energy diagram. This being said, it is very difficult to understand all the information contained in Figure 4 unless more description and guidance is provided.

Reviewer #1 (Remarks to the Author):

We are grateful to the reviewer for the thoughtful comments, which have enabled us to improve the manuscript. In the following, we provide a point-by-point response.

I'm sorry to say that I'm not excited about that paper. It deals with an old problem, i.e. the relaxation of the aqueous electron after photo-ionization of water. The experimental observation of a transient species in the NIR per se is not new, as the authors admit. It probably started with Ref 23 from 1997, where similar observations have been made. When I understand the authors correctly, what is supposed to be new is the interpretation of that transient species. They claim that it is a p-like wavefunction, rather than a continuously solvating s-like state. I however don't see how their new results would support that conclusion, i.e. in what sense a continuously moving absorption band contradicts their observation, while a discrete intermediate is in agreement. Independent of that, I'm pretty sure that p-like wavefunctions have been put forward before to explain this observation.

The electronic relaxation dynamics of ionized liquid water is indeed an "old problem", as the reviewer rightly points out. However, there are several limitations associated with the previous studies. *First*, the ~ 0.3 -ps laser pulses that were employed in the early studies offer lower time resolution, resulting in the large uncertainties in the formation (110 – 300 fs) and decay (240 – 545 fs) times of the intermediate state that was invoked to explain the observed dynamics (see p. 5 of the original manuscript). *Second*, the use of relatively long laser pulses for driving ionization allow the hydrated electron that is produced by the leading edge of the laser pulse to be photoexcited by the trailing edge of the laser pulse, as pointed out in ref. ¹, thus further complicating analysis. *Third*, the use of few-millimeter-thick sample targets housed within cuvettes in previous measurements inevitably give rise to cross-phase modulation artifacts that can be misinterpreted as the ultrafast response of the sample, as shown in ref. ². Moreover, the group velocity mismatch between the pump and probe pulses further degrades the time resolution. *Fourth*, some of the most recent measurements, performed with improved ~ 0.1 -ps time resolution, albeit with a narrower probe spectral range ($\sim 0.5 - 1 \mu\text{m}$), concluded that it is unnecessary to invoke the presence of a transient intermediate state, suggesting that the existence of such an intermediate state remains controversial; see, for example, refs. ^{1,3} and ⁴. These shortcomings, which have prevented a unified picture of the electronic relaxation dynamics of ionized liquid water from emerging, are now discussed on p. 2 – 3 of the revised manuscript. **(REV 1.1)**

In this study, we bring to bear on this "old problem" sub-two-cycle laser pulses spanning the visible to near-infrared ($0.5 - 0.9 \mu\text{m}$) and the short-wave infrared (SWIR, $1.1 - 1.7 \mu\text{m}$), made possible by recent advances in femtosecond laser technology. Our experimental data, along with the results obtained from theoretical simulations, enables us to reconstruct the most comprehensive picture of the ultrafast electronic relaxation dynamics of ionized liquid water to date. Our experimental results unambiguously resolve the sub-100-fs formation and subsequent decay of a pronounced SWIR absorption band. *The transient appearance of this SWIR absorption band at early time, on top of the continuous spectral shift (alluded to by the reviewer), is the experimental evidence for the existence of a transient electronic intermediate.* Direct relaxation from the conduction band to the hydrated electron ground state, on the other hand, would manifest itself as a gradual appearance of the hydrated electron absorption with a time constant of ~ 0.3 ps (following the lifetime of the conduction-band electron), at odds with the experimentally observed prompt appearance of the SWIR

Fig. R1. Comparison of the simulated time-resolved differential absorption spectra when the conduction-band electron relaxes **(a)** via the p state transient intermediate to the s state and **(b)** directly to the s state. The color scale is the differential absorption signal in arbitrary units. Note that both plots employ the same color scale. It is evident that the appearance of a SWIR absorption at early time delays, on top of the continuous spectral blue shift, is a signature of the transient intermediate.

with a time constant of ~ 60 fs. We now state this explicitly on p. 6 – 7 of the revised manuscript. **(REV 1.2)**

To further convince the reviewer that our experimental results support the existence of an intermediate state and that our ~ 10 -fs time resolution greatly facilitates its observation, we first show the time-resolved differential absorption spectra in two limits, with (Fig. R1a) and without (Fig. R1b) the involvement of the transient electronic intermediate. These contour plots are reconstructed based on our experimentally determined parameters, but with the probability of direct relaxation from the conduction band (CB) to the s state set to $P_{\text{dir}} = 0$ and $P_{\text{dir}} = 1$ in Figs. R1a and R1b, respectively. It is apparent that the appearance of the SWIR absorption band is a signature of the transient population of the intermediate state. Probing over a limited spectral range, as done in refs. ^{1,3} and ⁴, prevents the observation of the SWIR absorption, leading the authors of those works to conclude that the transient electronic intermediate does not exist. On the other hand, poor time resolution smears out the SWIR absorption signal, even if one does not consider additional complications associated with coherent artifacts and sequential photoexcitation within the same laser pulse, thereby preventing the accurate quantitative analysis of the time traces. In Figs. R2a and R2b, we show the reconstructed data from Fig. R1a convolved with instrument response functions characterized by FWHM values of 10 fs and 300 fs, respectively. It is evident that the lower signal amplitude and the broadening of the signal in time, observed in Fig. R2b, pose challenges to the accurate retrieval of the time constants, thereby explaining the large variance in the lifetimes of the intermediate state reported in the literature. These figures have been added to Section 6 of the revised Supplementary Information. **(REV 1.3)**

While numerous studies found no need to invoke the existence of the intermediate, those that did variously referred to it as the “prehydrated electron” (ref. ⁵), the “wet electron” (refs. ^{2,6-8} and), or the “weakly bound electron” (ref. ⁹). While the p state has indeed been put forth as the transient intermediate, e.g., see refs. ⁶ and ², we note that these assignments were made in the absence of supporting experimental evidence. On the other hand, our study resolves the lifetime of the transient intermediate as well as the isotope dependence of the lifetime, both of which are consistent with the characteristics of the p state deduced by the ultrafast

Fig. R2. Comparison of the simulated time-resolved differential absorption spectra when the conduction-band electron relaxes via the p state to the s state, as observed with a Gaussian FWHM instrument response function of (a) 10 fs and (b) 300 fs. The color scale is the differential absorption signal in arbitrary units. Note that both plots employ the same color scale. It is evident that the observation of the SWIR absorption at early time delays is hampered when the time resolution is ~ 300 fs, even in the absence of coherent artifacts.

spectroscopy of the equilibrium hydrated electron (refs. ¹⁰⁻¹⁴), thus providing the first direct evidence for the assignment of the transient intermediate state to the p electron. We have added this explanation to p. 8 – 9 of the revised manuscript. **(REV 1.4)**

To further support their conclusion, the present “NAMD simulations”. First, the level of theory is in essence not described in the main paper, but that would be very crucial as it has been shown in the past that the properties of the solvated electron very strongly depend on the level of description. Second, I don’t understand how the assignment is done. The authors refer to Fig.3b, which shows a red and a blue lobe, which are called charge density in the figure caption. The two lobes of a p -orbital have opposite sign, but that is NOT charge density. I must misunderstand something completely.

We thank the reviewer for this comment. Indeed, it is important to specify accurately the level of theory. We combine NAMD with ab initio real-time TDDFT, and we use the PBE0 functional with the fraction of the Hartree-Fock exchange increased to 40%, as established in the prior theoretical studies. We stated this and provided relevant references in the Supporting Information of the original manuscript. We did not adjust any parameters during our simulations, beyond what has been done in the published work. During the revision, we specified these computational details in the Methods section of the main text. **(REV 1.5)**

Fig. 3b shows the wave function (orbital), not charge density. We thank the reviewer for catching this error. We have changed the caption accordingly. **(REV 1.6)**

In conclusion, the paper needs quite some attention to work out better the arguments. Independent of that, after 25 years of work on this question, I would no longer consider it the most exciting science.

While it is true that the electronic relaxation dynamics of ionized liquid water has been investigated over a couple of decades, we believe that our unprecedented experimental time resolution allows us to reconstruct the most comprehensive picture for the electronic relaxation dynamics of ionized liquid water to date, thus shedding new insight into this “old problem”. Further, the reported ab initio NAMD simulations are the first of a kind.

Previously, NAMD simulations of solvated electrons have been performed at a much simpler, pseudopotential level of electronic structure theory. Conversely, ab initio studies of solvated electrons did not model NA excited state dynamics. The current simulation is first to perform NAMD of the hydrated electron at the ab initio (DFT) level of electronic structure theory. To stress the theoretical novelty, we emphasize this fact on p. 7 of the revised manuscript. (REV 1.7)

Reviewer #2 (Remarks to the Author):

The manuscript entitled "Observation of a transient intermediate in the ultrafast relaxation dynamics of the excess electron in ionized liquid water" by Low et al. (performed at the laboratory of prof. Loh and accompanied by theory calculations by Oleg Prezhdo) focuses on the electron formation upon multiphoton ionization or maybe excitation of liquid water. Even sixty years after its observation during pulse radiolysis, the hydrated electron remains to be somewhat enigmatic. For example, in 2010, the group of Benjamin Schwartz provocatively suggested a radically different view of the hydrated electron structure. The Science paper brought a huge excitement, with many research groups demonstrating that the outcome was just an artifact of poor electronic structure description. I make this lengthy historical introduction to emphasize that the general attractiveness of the subject should not compromise the attempt to understand in detail all the aspects of the theory and experiment. Below, I summarize some aspects I did not understand – and I might be easily wrong. Yet I would be happy to read the author's reaction.

We are grateful to the reviewer for the thoughtful comments for giving us the opportunity to respond to them. We believe that the manuscript has been improved following our addressing of the comments. In the following, we provide a point-by-point response.

My first question is whether the authors are sure what states are formed initially during the radiolysis. The strong field used in the work can electronically excite the molecule as well as ionize it. The excitation seems to lead to the observation of the very same hydrated electron. The authors should discuss in detail this aspect.

We thank the reviewer for this comment. In this work, we employ intense, few-cycle laser pulses to strong-field ionize liquid water. Due to the use of these intense, few-cycle pulses and the occurrence of strong-field ionization on the sub-cycle timescale, we believe that the electron is injected vertically into the conduction band (CB). An earlier study that probed the THz response of the CB electron, prepared by the strong-field ionization of liquid water, arrived at the same conclusion (ref. ¹⁵). As such, it is conceivable that strong-field ionization leads to the same hydrated electron species that would be produced by radiolysis. In other words, the electronic relaxation processes that are observed in this work are also applicable to those induced by conventional radiolysis. We have clarified this on p. 4 of the revised manuscript. **(REV 2.1)**

Relatively recently, the same process was investigated by time-resolved photoelectron spectroscopy in the laboratory of Hans-Jakob Wörner at ETH (see <https://www.science.org/doi/10.1126/sciadv.aaz0385>). TRPES is generally assumed to provide the most direct insight into the electronic structure, with both the energetics and anisotropy revealing the details of the dynamics. The initially formed electronic states seem to be much better selected in this case, demonstrating that both the excitation and ionization lead to the same product on a similar time scale. The interpretation is, naturally, different for both processes. The authors do not seem even aware of this work yet it seems to be an important anchor to their new study on a multiphoton generation of excess electrons. The work might be not fully relevant as it is done on clusters that are arguably solid-like. But is the difference large enough to ignore the comparison?

We thank the reviewer for this comment. We are aware of the said work by the Wörner group¹⁶. That work reported the observation of ultrafast proton/hydrogen atom transfer

dynamics and electron hydration dynamics following either 15.5-eV ionization or 9.3-eV excitation of water clusters, $(\text{H}_2\text{O})_N^+$ ($N \sim 400$). As the reviewer had guessed, we did not reference this work in our original manuscript because our results are not directly comparable to those obtained from water clusters due to their finite size, as we will explain below, as well as the possible ice-like nature of these large water clusters (it has been shown that water clusters with $N \gtrsim 275$ exhibit ice-like behavior [DOI: 10.1126/science.1225468]). We now see the omission of the work by the Wörner group as a shortcoming that we would like to address in the revised manuscript. This work is cited as ref. 33 in the revised manuscript.

The investigation by the Wörner group did not find any evidence for the existence of a transient intermediate state. The measured photoelectron asymmetry parameter, found to be independent of pump-probe time delay, is suggestive of a hydrated electron with an isotropic angular distribution. From there, it was inferred that the electron produced by ionization relaxes directly to the electronic ground state of the hydrated electron, albeit in a vibrationally excited state, without passing through the intermediate p state. We consider a few factors that could explain the discrepancy between their result and ours. *First*, unlike the non-vanishing anisotropy parameter that was previously observed for the photoexcited hydrated electron, in which orbital alignment was imprinted by photoexcitation of the hydrated electron with linearly polarized light (ref. ¹³), any p state that is produced in Wörner's experiment would have originated from relaxation of the conduction-band electron, during which orbital dealignment could occur. Loss of orbital alignment then yields the observed isotropic angular distribution. *Second*, our use of strong-field ionization favors the injection of an electron vertically into the CB. This is in contrast with Wörner's group use of 9.3-eV (133-nm) photoexcitation, which lies below the energy threshold required to inject an electron vertically into the CB. In that case, the observed ultrafast dynamics is explained in terms of H_2O undergoing sub-50-fs photodissociation to yield the hydronium radical (H_3O^\cdot), which then autoionizes to yield H_3O^+ and the s^* electron, thus circumventing the p state entirely. *Third*, in the experiments by the Wörner group that employ 15.5-eV (80-nm) vertical ionization, we note that the expected ejection length of $\langle r_0 \rangle \sim 3.8 \text{ nm}^{17}$ exceeds the ~ 1.4 -nm radius of the water cluster. As such, it is conceivable that the hydrated electron and its precursor, if any, are initially localized on the surface of the cluster. The abundance of surface defects and the low solvent reorganization energy requirements further favor the formation of a hydrated electron at the surface¹⁸. Considering that a surface-bound, ground-state hydrated electron has a low binding energy of $\sim 1.6 \text{ eV}^{18}$ or less¹⁹, and that its internalization and hence, concomitant increase in binding energy, occurs on a timescale of $\sim 0.5 \text{ ps}^{19}$, the p excited state of the hydrated electron might not even exist as a bound state on the surface if we assume that the $s - p$ energy gap on the surface is unchanged from the bulk value of 1.7 eV. Under such circumstances, one would expect the p state to be energetically inaccessible during the electronic relaxation of ionized water clusters, i.e., the conduction-band electron relaxes directly to the s^* state. *Fourth*, we note that their experimental time resolution of $\sim 85 \text{ fs}$ FWHM is lower than the ~ 10 -fs time resolution available in the present study; as such, it is conceivable that the short-lived p state, even if it had been formed, might have eluded detection. We have added this comparison, which we believe provide a context for our work, to p. 13 – 14 of the revised manuscript. **(REV 2.2)**

It would be fair to place the strong field generation of the hydrated electron in the title – it is one of the aspects marking a distinction from previous papers.

We thank the reviewer for this suggestion. We have revised the title to “Observation of a transient intermediate in the ultrafast relaxation dynamics of the excess electron in strong-field-ionized liquid water”. (REV 2.3)

The experiments are accompanied by ab initio MD calculations. I found this part confusing. The theory is not described in the main text, the approach is not justified. The way how the time-resolved vibrational spectra were obtained needs for sure a deeper discussion. I think that Nature Communications is not particularly restrictive on the size of the manuscript and all details needed to reconstruct the simulations should be provided.

We thank the reviewer for asking us to provide simulation details in the main text. Indeed, the original version of the paper gave such details in the Supporting Information. During the revision, we included the key simulation details in the main text. The quantum dynamics simulation methodology combined NAMM for the description of quantum transitions within the electronic manifold and interactions between electrons and nuclei, with real-time TDDFT for evolution of the electronic subsystem. Such methodology has been used successfully with a broad range of condensed phase and molecular systems. We did use a larger-than-normal fraction of the Hartree-Fock exchange in the PBE0 DFT functional. This is needed for proper description of the hydrated electron properties, as established in prior theoretical works. We state this and cited these works in the Methods section, now added to the Methods section of the main text in the revised manuscript. (REV 2.4)

To obtain the time-resolved vibrational spectra, more specifically, the electron-phonon influence spectra, we computed Fourier transforms of the energy gaps between the initial and final states, i.e., between the lowest energy conduction band state and p state, and between the p and s states. To obtain time-resolution, the Fourier transforms are computed for the parts of the trajectories in which the electron is already relaxed to the initial state for each transition. Because these trajectory parts are relatively short, the spectral resolution is relatively low. Nevertheless, the influence spectra can clearly identify what kind of modes – translational, librational, internal bending – are participating in a particular transition because the frequencies of these motions differ by hundreds of cm^{-1} . We have added these details to the Methods section of the main text during the revision. (REV 2.5)

Some minor issues. The authors cite Sanche’s hypothesis on the role of pre-solvated electrons in DNA damage. It should be fair to mention that the community remains unconvinced (and I like the hypothesis).

We thank the reviewer for pointing this out. We have revised the last sentence of the first paragraph to read, “The electrons, on the other hand, are ~~known~~ postulated to induced genomic damage by dissociative electron attachment⁵.” (REV 2.6)

Figure 4 should be augmented, adding especially all recent photoemission data.

We thank the reviewer for this suggestion. We have added to the text that accompanies Fig. 4 a range of energetic parameters obtained from recent studies; please see p. 17 of the revised manuscript. (REV 2.7)

To conclude, I find the study interesting yet more work should be invested into placing the present finding in a context. The study could be also more “honest” in the sense that alternative hypotheses to explain the data are used.

We thank the reviewer for finding our work interesting. We believe that our discussion of the limitations encountered in earlier optical pump-probe measurements of ionized liquid water (REV 1.1) and the comparison to the recent work by the Wörner group on ionized water clusters (REV 2.2) provide more context for our findings. Beyond these, we have also inserted into p. 13 – 15 of the revised manuscript a brief discussion of the recent studies by the Signorell^{20,21} and Suzuki²² groups, performed on water clusters, $(\text{H}_2\text{O})_N^+$ ($\langle N \rangle \sim 300$), and liquid water, respectively. Neither study observed the p state intermediate, perhaps due to the limited $\sim 70 - 220$ fs time resolution of the experimental setups and coherent artifacts obfuscating the early-time dynamics. In addition, similar to the situation encountered in the study by the Wörner group, the cluster study by the Signorell group is complicated by the ejection length $\langle r_0 \rangle \sim 3.2$ nm being greater than the dimension of the water cluster, thus favoring initial localization of the electron on the surface, where the p state might be energetically inaccessible. (REV 2.8)

We thank the reviewer for encouraging us to explore possible alternative hypotheses to explain our data. In the revised manuscript, we now consider three additional possibilities. *First* is the direct population of the vibrationally hot s^* state from the CB. The corresponding simulated spectrally resolved transient absorption (see Fig. R1b under our response to Reviewer #1) is missing the prominent SWIR transient absorption that is clearly observed in the experiments. The appearance of the SWIR transient absorption at early time delays, beyond the simultaneous spectral shift and the growth of the s state absorption, necessitates the involvement of a transient intermediate in the kinetic model. The inclusion of the transient intermediate reproduces the SWIR absorption feature in the simulated time-resolved transient absorption spectra (see Fig. R1a). *Second*, beyond the p state, we also consider the involvement of the hypervalent hydronium radical species, H_3O^\cdot , as an intermediate state. In previous time-resolved photoelectron spectroscopy studies, the formation of s^* electrons following 9.3-eV photoexcitation of water clusters¹⁶ and 7.7-eV photoexcitation of liquid water²² to the electronically excited \tilde{A} state was attributed to the dissociation of H_2O to yield the H_3O^\cdot radical, which then autoionizes to yield H_3O^+ and the s^* electron, i.e.,

For water clusters, s^* electron formation via this channel was found occur on an overall timescale of 43 fs and 61 fs for H_2O and D_2O clusters, respectively. In our experiments, resonance-enhanced strong-field ionization via the \tilde{A} state could initiate these dynamics. However, while the formation of H_3O^\cdot as a transient intermediate is consistent with the kinetic scheme inferred from our experiments, we note that ab initio calculations predict a strong optical absorption signature²³ at ~ 780 nm for hydrated H_3O^\cdot , inconsistent with the experimentally observed absorption band at $\sim 1.6 \mu\text{m}$. Moreover, formation of H_3O^\cdot by the photodissociation of the water O–H bond is expected to occur on the sub-10-fs timescale if we consider the repulsive nature of the \tilde{A} state accessed by ~ 8 -eV photoexcitation^{24,25}. In contrast, the intermediate state is observed in our experiments to form on longer timescales. For these reasons, we exclude the possibility of identifying H_3O^\cdot as the transient intermediate. *Third*, instead of producing the conduction-band (CB) electron via vertical ionization, we consider the possibility that the \tilde{A} state autoionizes to yield the CB electron¹⁷, which then undergoes the sequential relaxation process outlined in the manuscript. Our experiments are unable to distinguish the formation of the CB electron via the autoionization or vertical ionization channels because they do not probe the spectroscopic observable of CB

electrons, located in the terahertz. Future experiments that probe the terahertz absorption of ionized liquid water can resolve the delayed appearance of CB electrons that is associated with the autoionization channel. We have added the above discussion to p. 12 – 13 of the revised manuscript. **(REV 2.9)**

Reviewer #3 (Remarks to the Author):

This is certainly a nice work providing some new insight into the ultrafast relaxation dynamics of the excess electron in ionized liquid water. Specifically, the authors suggest that the entire population of electrons injected into the water conduction band goes through a previously (unconfirmed) trap state in the band gap to subsequently form the equilibrated hydrated electron. I consider the new spectroscopic data, largely due to improved temporal resolution, and the accompanying theoretical computations, an important contribution in advancing our understanding of the respective ultrafast electronic relaxation processes, including their mechanistic details.

We thank the reviewer for considering our work an important contribution to advancing the understanding of the ultrafast electronic relaxation processes in ionized liquid water. At the same time, we are also grateful to the reviewer for the thoughtful comments, which have allowed us to improve the manuscript. In the following, we provide a point-by-point response.

However, the results need to be discussed in much greater depth which includes to thoroughly relate to a number of findings from previous works, not addressed in the present manuscript. With the extended discussion the manuscript won't fit format the communication format, also considering the many necessary assumptions (I come back to that) the authors need to make in order to reach at their interpretation of the results. As much as I appreciate the authors' overall work, there is room for alternative conclusions. Furthermore, I am quite disappointed how little the authors comment on water being a large-band-gap semiconductor and the implications in the present context. There are also several experimental aspects that need to be addressed more thoroughly. In the following I provide some specific comments.

The reviewer's comments about relating our results to previous findings and offering alternative interpretations of the results are well-taken. During the revision of the manuscript, we discuss critically the limitations encountered in the earlier studies, which prevented a consensus for the existence of the intermediate state from being established, and furthermore we put our work in a broader context by discussing our findings in relation to recent results obtained from the time-resolved photoelectron spectroscopy of ionized water clusters and liquid water. More details are given below.

Earlier studies of the electronic relaxation dynamics of ionized liquid water encountered several limitations. *First*, the ~ 0.3 -ps laser pulses that were employed in the early studies offer lower time resolution, resulting in the large uncertainties in the formation (110 – 300 fs) and decay (240 – 545 fs) times of the intermediate state that was invoked to explain the observed dynamics (see p. 5 of the original manuscript). *Second*, the use of relatively long laser pulses for driving ionization allow the hydrated electron that is produced by the leading edge of the laser pulse to be photoexcited by the trailing edge of the laser pulse, as pointed out in ref. ¹, thus further complicating analysis. *Third*, the use of few-millimeter-thick sample targets housed within cuvettes in previous measurements inevitably give rise to cross-phase modulation artifacts that can be misinterpreted as the ultrafast response of the sample, as shown in ref. ². Moreover, the group velocity mismatch between the pump and probe pulses further degrades the time resolution. *Fourth*, some of the most recent measurements, performed with improved ~ 0.1 -ps time resolution, albeit with a narrower probe spectral range ($\sim 0.5 - 1 \mu\text{m}$), concluded that it is unnecessary to invoke the presence of a transient intermediate state, suggesting that the existence of such an intermediate state remains

controversial; see, for example, refs. ^{1,3} and ⁴. These shortcomings, which have prevented a unified picture of the electronic relaxation dynamics of ionized liquid water from emerging, are now discussed on p. 2 – 3 of the revised manuscript. **(REV 3.1)**

On p. 13 – 15 of the revised manuscript, we compare our results to the recent report by the Wörner group on the ultrafast dynamics induced by the ionization of water, albeit in the form of clusters, $(\text{H}_2\text{O})_N^+$ ($N \sim 400$)¹⁶. Their investigation did not find any evidence for the existence of a transient intermediate state. The measured photoelectron asymmetry parameter, found to be independent of pump-probe time delay, is suggestive of a hydrated electron with an isotropic angular distribution. From there, it was inferred that the electron produced by ionization relaxes directly to the electronic ground state of the hydrated electron, albeit in a vibrationally excited state, without passing through the intermediate p state. We consider a few factors that could explain the discrepancy between their result and ours. *First*, unlike the non-vanishing anisotropy parameter that was previously observed for the photoexcited hydrated electron, in which orbital alignment was imprinted by photoexcitation of the hydrated electron with linearly polarized light (ref. ¹³), any p state that is produced in Wörner's experiment would have originated from relaxation of the conduction-band electron, during which orbital dealignment could occur. Loss of orbital alignment then yields the observed isotropic angular distribution. *Second*, our use of strong-field ionization leads to the injection of an electron vertically into the CB. This is in contrast with Wörner's group use of 9.3-eV (133-nm) photoexcitation, which lies below the energy threshold required to inject an electron vertically into the CB. In that case, the observed ultrafast dynamics is explained in terms of H_2O undergoing sub-50-fs photodissociation to yield the hydronium radical ($\text{H}_3\text{O}\cdot$), which then autoionizes to yield H_3O^+ and the s^* electron, thus circumventing the p state entirely. *Third*, in the experiments by the Wörner group that employ 15.5-eV (80-nm) vertical ionization, we note that the expected ejection length of $\langle r_0 \rangle \sim 3.8 \text{ nm}$ ¹⁷ exceeds the ~ 1.4 -nm radius of the water cluster. As such, it is conceivable that the hydrated electron and its precursor, if any, are initially localized on the surface of the cluster. The abundance of surface defects and the low solvent reorganization energy requirements further favor the formation of a hydrated electron at the surface¹⁸. Considering that a surface-bound, ground-state hydrated electron has a low binding energy of $\sim 1.6 \text{ eV}$ ¹⁸ or less¹⁹, and that its internalization and hence, concomitant increase in binding energy, occurs on a timescale of $\sim 0.5 \text{ ps}$ ¹⁹, the p excited state of the hydrated electron might not even exist as a bound state on the surface if we assume that the $s - p$ energy gap on the surface is unchanged from the bulk value of 1.7 eV. Under such circumstances, one would expect the p state to be energetically inaccessible during the electronic relaxation of ionized water clusters, i.e., the conduction-band electron relaxes directly to the s^* state. *Fourth*, we note that their experimental time resolution of $\sim 85 \text{ fs}$ FWHM is lower than the ~ 10 -fs time resolution available in the present study; as such, it is conceivable that the short-lived p state, even if it had been formed, might have eluded detection. Here, we also note the recent studies by the Signorell^{20,21} and Suzuki²² groups, performed on water clusters, $(\text{H}_2\text{O})_N^+$ ($N \sim 300$), and liquid water, respectively. Neither study observed the p state intermediate, perhaps due to the limited $\sim 70 - 220 \text{ fs}$ time resolution of the experimental setups and coherent artifacts obfuscating the early-time dynamics. In addition, similar to the situation encountered in the study by the Wörner group, the cluster study by the Signorell group is complicated by the ejection length $\langle r_0 \rangle \sim 3.2 \text{ nm}$ being greater than the average dimension of the water cluster ($\sim 1.3 \text{ nm}$ radius), thus favoring initial localization of the electron on the surface, where the p state might be energetically inaccessible. **(REV 3.2)**

On p. 12 – 13 of the revised manuscript, we now consider three alternative interpretations of the experimental data. *First* is the direct population of the vibrationally hot s^* state from the CB. The corresponding simulated spectrally resolved transient absorption (see Fig. R1b under our response to Reviewer #1) is missing the prominent SWIR transient absorption that is observed in the experiments. The appearance of the SWIR transient absorption at early time delays, beyond the simultaneous spectral shift and the growth of the s state absorption, necessitates the involvement of a transient intermediate in the kinetic model. The inclusion of the transient intermediate reproduces the SWIR absorption feature in the simulated time-resolved transient absorption spectra (see Fig. R1a). *Second*, beyond the p state, we also consider the involvement of the hypervalent hydronium radical species, H_3O^\cdot , as an intermediate state. In previous time-resolved photoelectron spectroscopy studies, the formation of s^* electrons following 9.3-eV photoexcitation of water clusters¹⁶ and 7.7-eV photoexcitation liquid water²² to the electronically excited \tilde{A} state was attributed to the dissociation of H_2O to yield the H_3O^\cdot radical, which then autoionizes to yield H_3O^+ and the s^* electron, i.e.,

For water clusters, s^* electron formation via this channel was found occur on an overall timescale of 43 fs and 61 fs for H_2O and D_2O clusters, respectively. In our experiments, resonance-enhanced strong-field ionization via the \tilde{A} state could initiate these dynamics. However, while the formation of H_3O^\cdot as a transient intermediate is consistent with the kinetic scheme inferred from our experiments, we note that ab initio calculations predict a strong optical absorption signature²³ at ~ 780 nm for hydrated H_3O^\cdot [Domcke PCCP 2002], inconsistent with the experimentally observed absorption band at $\sim 1.6 \mu\text{m}$. Moreover, formation of H_3O^\cdot by the photodissociation of the water O–H bond is expected to occur on the sub-10-fs timescale, given the repulsive nature of the \tilde{A} state accessed by ~ 8 -eV photoexcitation^{24,25}. In contrast, the intermediate state is observed in our experiments to form on longer timescales. For these reasons, we exclude the possibility of identifying H_3O^\cdot as the transient intermediate. *Third*, instead of producing the conduction-band (CB) electron via vertical ionization, we consider the possibility that the \tilde{A} state autoionizes to yield the CB electron¹⁷, which then undergoes the sequential relaxation process outlined in the manuscript. Our experiments are unable to distinguish the formation of the CB electron via the autoionization or vertical ionization channels because they do not probe the spectroscopic observable of CB electrons, located in the terahertz. Future experiments that probe the terahertz absorption of ionized liquid water can resolve the delayed appearance of CB electrons that is associated with the autoionization channel. **(REV 3.3)**

Please see below for our response to the point that the reviewer had raised about water being a large band-gap semiconductor.

Pages 3,4: Regarding the multi-(1.96 eV) photon ionization, I wonder how well the order of the process can be controlled. My concern is that ‘ionization’ is not limited to the injection of electrons deep into the CB but also produces true photoelectrons, emitted into vacuum. This additional channel would presumably affect the interpretation of the data, and should be commented on. A related issue is if strong-field ionization can be considered to be equivalent to single-photon ionization with regard to the relaxation dynamics of the excess electron. To my understanding this is not obvious at all, and there may be a possibility that this has a

signature in the experimental data.

We thank the reviewer for this comment. Our pump fluence dependence measurements (see Section 3 of the Supplementary Information) give photon orders that are very close to four (4.07 ± 0.07 with visible probing and 3.99 ± 0.01 with SWIR probing). This result indicates that a four-photon resonance enhanced multiphoton ionization process dominates strong-field ionization. As the reviewer points out, strong-field ionization also leads to the ejection of electrons from the bulk liquid. However, these photoelectrons, detected in photoemission experiments, do not form hydrated electrons and therefore do not contribute to our transient absorption signal, which probe the electron dynamics within the bulk liquid. We have clarified this on p. 4 – 5 of the revised manuscript. **(REV 3.4)**

The reviewer also asks if strong-field ionization and single-photon ionization can be considered to be equivalent. In this work, we employ intense, few-cycle laser pulses to strong-field ionize liquid water. Due to the use of these intense, few-cycle pulses and the occurrence of strong-field ionization on the sub-cycle timescale, we believe that the electron is injected vertically into the CB. (An earlier study that probed the THz response of the CB electron, prepared by the strong-field ionization of liquid water, arrived at the same conclusion¹⁵.) As such, we believe that strong-field ionization and single-photon ionization are equivalent in the sense that both involve vertical transitions. This has already been clarified on p. 3 – 4 of the original manuscript.

Page 5: The authors say: “Moreover, the absorption maxima at $\sim 1.3 \mu\text{m}$ and $\sim 0.85 \mu\text{m}$, extracted previously from global fitting of absorption spectra, are not observed in the present work.” Is that a correct statement? There seems to be intensity near $1.3 \mu\text{m}$, and the $0.85 \mu\text{m}$ range is not covered in Figure 2. The authors should explain the reason for this discrepancy (if real).

We thank the reviewer for this comment. We would like to point out that the sentence in the manuscript pertains to the location of the absorption maxima. While our data shows that the SWIR absorption is broad, with significant transient absorption at $1.3 \mu\text{m}$, the absorption maximum clearly resides at longer wavelengths ($\sim 1.6 \mu\text{m}$). We would also like to point out that $0.85 \mu\text{m}$ is covered in Fig. 2a (the horizontal axis spans 600 – 900 nm).

On the same page it also says: “...the use of $\sim 8 - 9 \text{ eV}$ for photoionization, below the $\sim 10\text{-eV}$ energy for vertically injecting eCB into liquid water^{26,27} ...” This is the point where I wished the authors attempted to consider liquid water as a semiconductor, and quantitatively relate to the size of water’s band gap. This should be inspired by works of the Sprik (C. Adriaanse et al., JPCL 3, 3411, 2012) and Galli (A. Gaiduk et al., Nat Comm 9, 247, 2018) groups. Furthermore, as I had pointed out above, 10 eV photon energy corresponds also to the onset of producing true photoelectrons. How are these processes related, and how do they potentially affect each other? Wouldn’t it be useful to explicitly address and distinguish these two channels? In fact, even the title of this manuscript can be misunderstood.

We thank the reviewer for this comment and for referring us to the works by Sprik and Galli, now cited in the revised manuscript as refs. 42 and 43, respectively. The reviewer seeks to make a connection between our measurement results and the large band gap of liquid water. With liquid water at its equilibrium geometry, the (vertical) band gap is $\sim 10 - 11 \text{ eV}$; these values are obtained by considering the recently reported vertical ionization potential (11.33 eV)^{26,27} and the various computed vertical electron affinities ($0.2, 0.74, 0.97, 1.1 \text{ eV}$)²⁸⁻³¹ of

liquid water. Allowing for solvent reorganization leads to a reduced (adiabatic) band gap of ~ 7 eV^{32,33}. While our experiments do not measure these energetic parameters – our experiments probe the relaxation from the CB to the various hydrated electron states (in the terminology of semiconductor physics, these states would correspond to defect levels in the band gap) – they nevertheless determine the accessible ionization mechanisms. When the total energy deposited by the pump pulse exceeds the vertical band gap, the electron is injected vertically into the CB. When the total energy deposited by the pump pulse falls between the vertical and adiabatic band gaps, however, the CB electron is produced via autoionization, which involves solvent reorganization and therefore a possible change in the electronic relaxation timescales. On p. 13 of the revised manuscript, we explain these ionization mechanisms in relation to the vertical and adiabatic band gaps of liquid water.

(REV 3.5)

The ejection of photoelectrons does not affect the experimental results, as we had explained above in relation to an earlier comment from this reviewer (please see REV 3.1), because our measurements probe the transient absorption of excess electrons in the bulk liquid, independent of photoelectrons that are ejected from the liquid jet. Moreover, our use of the term “ionization” is consistent with the literature, where the removal of an electron from a water molecule, regardless of whether the electron is injected into the CB or into vacuum (as a photoelectron), is referred to as “ionization”; please see, for example, refs. ^{1,4,5,34,35}. We also note that the appearance of “excess electron” in the title should clarify that the electron is injected into the CB.

Page 7: It says: “The observed $\tau_{\text{int}} \sim \tau_p$ and their similar isotope dependence strongly suggest that e_{int} corresponds to e_p , i.e., e_p exists as an intermediate state in the electronic relaxation of ionized liquid water.” I find this a very important result which should be discussed in greater detail though, expanding on the role of H₂O⁺, and certainly that text should not be moved to the SI but rather be part of the Discussion section. Furthermore, I believe that the concept of a double-cavity structure is new, and the authors should emphasize this aspect further.

We thank the reviewer for this comment and for appreciating the importance of our result, showing that the p state exists as an intermediate state in the electronic relaxation of ionized liquid water. Because the excited-state absorption spectrum of the p state electron is obtained from a three-pulse (ionization-excitation-probe) experiment, different from the two-pulse (ionization-probe) experiment employed in the bulk of our studies, we have elected to place the transient absorption spectrum in the Supplementary Information. Following the advice of the reviewer, we have transferred most of the text on the excited-state absorption of the hydrated electron in the SWIR from Section 6 of the Supplementary Information to p. 8 of the main text. **(REV 3.6)**

The advice by the reviewer to expand upon the discussion associated with our assignment of the intermediate state to the p electron is well-taken. Accordingly, we have added the following discussion to p. 15 – 16 of the revised manuscript. *First*, we note that our experiments on ionized liquid water and other experiments on the excited-state dynamics of the hydrated electron involve different initial solvent configurations. In the case of ionized liquid water, the initial solvent configuration is that of liquid water at equilibrium. In the case of the excited-state dynamics of the hydrated electron, however, the hydrated electron initially resides in a solvent cavity of dimensions comparable to the radius of gyration of the hydrated electron (2.44 Å)³⁶. Hence, our assignment of the intermediate state to the p state

suggests that solvent reorganization within the timescale for relaxation from the CB (~ 0.3 ps) yields the solvent configuration of the hydrated electron. This is supported by the ab initio NAMD simulations that show relaxation of the high energy excited electron from deep inside in the CB to the p shaped state exhibiting two lobes (Fig. 3b). *Second*, unlike the experiments on the excited-state dynamics of the hydrated electron, where the observed ultrafast dynamics is associated solely with the hydrated electron, ionized liquid water simultaneously exhibits both electron and hole dynamics. Therefore, while our present study focuses on the electronic relaxation dynamics, it is important to realize that the valence hole created by ionization undergoes ultrafast temporal evolution at the same time^{35,37-39}. According to ab initio molecular dynamics simulations, the initially delocalized valence hole localizes on the ~ 30 -fs timescale onto a single water molecule, forming the $\text{H}_2\text{O}^{\cdot+}$ radical cation^{35,37}, which subsequently undergoes proton transfer to a neighboring water molecule to produce the hydronium ion, H_3O^+ , and the hydroxyl radical, OH^{\cdot} . A recent femtosecond soft X-ray absorption study found a timescale of ~ 50 fs for this ultrafast proton transfer reaction³⁷. An obvious question that arises is the extent to which electronic relaxation dynamics is affected by hole localization and subsequent proton transfer. The latter is accompanied by the contraction of the intermolecular $\text{O}\cdots\text{O}$ distance between $\text{H}_2\text{O}^{\cdot+}$ and the neighboring H_2O molecules in its immediate vicinity. The contraction of the $\text{O}\cdots\text{O}$ bond (to 2.4 Å) promotes proton transfer, whereupon the $\text{O}\cdots\text{O}$ distance returns to that of equilibrium neutral water (2.7 Å). It is conceivable that the valence hole-induced solvent reorganization dynamics could interfere with the electronic relaxation dynamics. However, we note that our ab initio nonadiabatic molecular dynamics simulations reproduce the timescales for the electronic relaxation dynamics even though they do not consider the competing hole dynamics. Moreover, an ejection length of ~ 40 Å, determined from a recent time-resolved THz study of strong-field-ionized liquid water¹⁵, implies that the initial ionization site and the site at which the electron localizes to form the p state is separated by a distance of ~ 16 solvent shells. The large electron-hole separation produced by strong-field ionization provides a plausible explanation for the absence of any interference between the electron and hole dynamics.

(REV 3.7)

The ab initio NAMD simulations demonstrate that the p state contains two lobes, as expected, and that the two lobes occupy two adjacent cavities. The earlier NAMD simulations based on a single-particle pseudopotential description of the hydrated electron showed that the p state occupied an elongated cavity that turned into a spherical cavity upon relaxation to the s state⁴⁰. The current ab initio DFT level of theory, which includes not only the extra electron but also the valence electrons of all the water molecules, produces a somewhat different structure. The two lobes of the p state are separated by a low electron density region that allows a few water molecules to penetrate between the lobes. This double-cavity type structure occurs because the p state has a node in the middle, and in the absence of electron density at the node, there is no Pauli repulsion that pushes electrons of water molecules away from this region. The pseudopotential approach employed in earlier studies either creates an elongated single cavity⁴⁰ or no cavity⁴¹, depending on the pseudopotential parameters. The more sophisticated ab initio description that includes all the valence electrons allows the double-cavity to be formed, with water molecules pushed away from the regions of high p electron density and allowing water molecules to penetrate regions of low p electron density near the node. We have added this point to p. 11 of the revised manuscript. **(REV 3.8)**

Pages 9,10: I am quite puzzled by the energy values that were chosen on page 9 (bottom), and then used to construct Figure 4. Specifically, why have the authors chosen $\text{VDE} = 3.76$ eV and not lower values? But the bigger worry is the $\text{V}_0 = 0.97$ eV value. Admittedly, this value

has been debated for a long time. Not mentioning that it is likely much smaller (Gaiduk et al.) is not helpful since there are alternatives to construct such energy diagram. This being said, it is very difficult to understand all the information contained in Figure 4 unless more description and guidance is provided.

We thank the reviewer for this comment. Amongst the parameters that are extracted from the literature and presented in Fig. 4 – the vertical electron affinity (V_0), adiabatic electron affinity (AEA), vertical detachment energy (VDE), and transition energy of the hydrated electron (E_s) – there is greatest degree of uncertainty surrounding VDE and V_0 . (Please also note that the estimation of the energetic position of e^\dagger does not require knowledge of AEA.) The VDE value that we have chosen is from a recent study in which extreme ultraviolet (EUV) probe pulses were employed⁴². Under these conditions, it is found that the measured photoelectron kinetic energy distributions are relatively insensitive to inelastic scattering within the liquid, thereby enabling the accurate retrieval of the VDE (3.76 ± 0.05 eV). We also note that the VDE obtained from EUV probing is comparable to that obtained from an earlier study that employed multi-wavelength UV probing with energy-dependent corrections for electron scattering (3.7 ± 0.1 eV)^{21,43}. Without corrections for electron scattering, the measured VDEs are found to be generally lower, in the range of 3.2 – 3.7 eV⁴⁴⁻⁴⁷.

As alluded to by the reviewer, a definitive value for V_0 has remained elusive to date. In our original manuscript, we chose to use the value $V_0 = 0.97$ eV, obtained from thermodynamic integration using hybrid functionals without the inclusion of nuclear quantum effects³⁰. We thank the reviewer for drawing our attention to the important work by Gaiduk et al.²⁸, which combined path-integral molecular dynamics with non-self-consistent G_0W_0 calculations starting from hybrid functional wave functions to yield V_0 of 0.2 eV. In the revised manuscript, we consider these V_0 values as well as that from a follow-up work by Ziaei and Bredow (1.1 eV)³¹, performed at the self-consistent GW level of theory with implicit vertex corrections, albeit on a smaller simulation cell (32 vs. 64 water molecules used by Gaiduk). On p. 17 – 18 of the revised manuscript, we now cite V_0 obtained from these various studies and note that its value has been a subject of debate. **(REV 3.10)** Furthermore, on p. 16 – 17 of the revised manuscript, we now provide definitions of the various parameters that appear in Fig. 4. **(REV 3.11)**

References

1. Vilchiz V. H., Kloepfer J. A., Germaine A. C., Lenchenkov V. A., Bradforth S. E. Map for the relaxation dynamics of hot photoelectrons injected into liquid water *via* anion threshold photodetachment and above threshold solvent ionization. *J. Phys. Chem. A* **105**, 1711-1723 (2001).
2. Wang C.-R., Luo T., Lu Q.-B. On the lifetimes and physical nature of incompletely relaxed electrons in liquid water. *Phys. Chem. Chem. Phys.* **10**, 4463-4470 (2008).
3. Hertwig A., Hippler H., Unterreiner A.-N. Transient spectra, formation, and geminate recombination of solvated electrons in pure water UV-photolysis: An alternative view. *Phys. Chem. Chem. Phys.* **1**, 5633-5642 (1999).
4. Kambhampati P., Son D. H., Kee T. W., Barbara P. F. Solvation dynamics of the hydrated electron depends on its initial degree of electron delocalization. *J. Phys. Chem. A* **106**, 2374-2378 (2002).
5. Migus A., Gauduel Y., Martin J. L., Antonetti A. Excess electrons in liquid water: First evidence of a prehydrated state with femtosecond lifetime. *Phys. Rev. Lett.* **58**, 1559-1562 (1987).
6. Long F. H., Lu H., Eisenthal K. B. Femtosecond studies of the presolvated electron: An excited state of the solvated electron? *Phys. Rev. Lett.* **64**, 1469-1472 (1990).
7. Shi X., Long F. H., Lu H., Eisenthal K. B. Femtosecond electron solvation kinetics in water. *J. Phys. Chem.* **100**, 11903-11906 (1996).
8. Laenen R., Roth T., Laubereau A. Novel precursors of solvated electrons in water: Evidence for a charge transfer process. *Phys. Rev. Lett.* **85**, 50-53 (2000).
9. Pépin C., Goulet T., Houde D., Jay-Gerin J. P. Observation of a continuous spectral shift in the solvation kinetics of electrons in neat liquid deuterated water. *J. Phys. Chem. A* **101**, 4351-4360 (1997).
10. Pshenichnikov M. S., Baltuška A., Wiersma D. A. Hydrated-electron population dynamics. *Chem. Phys. Lett.* **389**, 171-175 (2004).
11. Elkins M. H., Williams H. L., Shreve A. T., Neumark D. M. Relaxation mechanism of the hydrated electron. *Science* **342**, 1496-1499 (2013).
12. Elkins M. H., Williams H. L., Neumark D. M. Isotope effect on hydrated electron relaxation dynamics studied with time-resolved liquid jet photoelectron spectroscopy. *J. Chem. Phys.* **144**, 184503 (2016).
13. Karashima S., Yamamoto Y.-i., Suzuki T. Resolving nonadiabatic dynamics of hydrated electrons using ultrafast photoemission anisotropy. *Phys. Rev. Lett.* **116**, 137601 (2016).
14. Karashima S., Yamamoto Y.-i., Suzuki T. Ultrafast internal conversion and solvation of electrons in water, methanol, and ethanol. *J. Phys. Chem. Lett.* **10**, 4499-4504 (2019).
15. Savolainen J., Uhlig F., Ahmed S., Hamm P., Jungwirth P. Direct observation of the collapse of the delocalized excess electron in water. *Nat. Chem.* **6**, 697-701 (2014).
16. Svoboda V., Michiels R., LaForge A. C., Med J., Stienkemeier F., *et al.* Real-time observation of water radiolysis and hydrated electron formation induced by extreme-ultraviolet pulses. *Sci. Adv.* **6**, eaaz0385 (2020).

17. Elles C. G., Jailaubekov A. E., Crowell R. A., Bradforth S. E. Excitation-energy dependence of the mechanism for two-photon ionization of liquid H₂O and D₂O from 8.3 to 12.4 eV. *J. Chem. Phys.* **125**, 044515 (2006).
18. Verlet J. R. R., Bragg A. E., Kammrath A., Cheshnovsky O., Neumark D. M. Observation of large water-cluster anions with surface-bound excess electrons. *Science* **307**, 93-96 (2005).
19. Coons M. P., You Z.-Q., Herbert J. M. The hydrated electron at the surface of neat liquid water appears to be indistinguishable from the bulk species. *J. Am. Chem. Soc.* **138**, 10879-10886 (2016).
20. Ban L., West C. W., Chasovskikh E., Gartmann T. E., Yoder B. L., *et al.* Below band gap formation of solvated electrons in neutral water clusters? *J. Phys. Chem. A* **124**, 7959-7965 (2020).
21. Gartmann T. E., Ban L., Yoder B. L., Hartweg S., Chasovskikh E., *et al.* Relaxation dynamics and genuine properties of the solvated electron in neutral water clusters. *J. Phys. Chem. Lett.* **10**, 4777-4782 (2019).
22. Yamamoto Y.-i., Suzuki T. Ultrafast dynamics of water radiolysis: Hydrated electron formation, solvation, recombination, and scavenging. *J. Phys. Chem. Lett.* **11**, 5510-5516 (2020).
23. Sobolewski A. L., Domcke W. Hydrated hydronium: A cluster model of the solvated electron? *Phys. Chem. Chem. Phys.* **4**, 4-10 (2002).
24. Zhang J., Imre D. G. Spectroscopy and photodissociation dynamics of H₂O: Time-dependent view. *J. Chem. Phys.* **90**, 1666-1676 (1989).
25. Trushin S. A., Schmid W. E., Fuß W. A time constant of 1.8 fs in the dissociation of water excited at 162 nm. *Chem. Phys. Lett.* **468**, 9-13 (2009).
26. Thürmer S., Malerz S., Trinter F., Hergenhausen U., Lee C., *et al.* Accurate vertical ionization energy and work function determinations of liquid water and aqueous solutions. *Chem. Sci.* **12**, 10558-10582 (2021).
27. Signorell R., Winter B. Photoionization of the aqueous phase: Clusters, droplets and liquid jets. *Phys. Chem. Chem. Phys.* **24**, 13438-13460 (2022).
28. Gaiduk A. P., Pham T. A., Govoni M., Paesani F., Galli G. Electron affinity of liquid water. *Nat. Commun.* **9**, 247 (2018).
29. Chen W., Ambrosio F., Miceli G., Pasquarello A. Ab initio electronic structure of liquid water. *Phys. Rev. Lett.* **117**, 186401 (2016).
30. Ambrosio F., Miceli G., Pasquarello A. Electronic levels of excess electrons in liquid water. *J. Phys. Chem. Lett.* **8**, 2055-2059 (2017).
31. Ziaei V., Bredow T. Probing ionization potential, electron affinity and self-energy effect on the spectral shape and exciton binding energy of quantum liquid water with self-consistent many-body perturbation theory and the Bethe–Salpeter equation. *J. Phys.: Condens. Matter* **30**, 215502 (2018).
32. Coe J. V., Earhart A. D., Cohen M. H., Hoffman G. J., Sarkas H. W., *et al.* Using cluster studies to approach the electronic structure of bulk water: Reassessing the vacuum level, conduction band edge, and band gap of water. *J. Chem. Phys.* **107**, 6023-6031 (1997).

33. Fang C., Li W.-F., Koster R. S., Klimeš J., van Blaaderen A., *et al.* The accurate calculation of the band gap of liquid water by means of GW corrections applied to plane-wave density functional theory molecular dynamics simulations. *Phys. Chem. Chem. Phys.* **17**, 365-375 (2015).
34. Crowell R. A., Bartels D. M. Multiphoton ionization of liquid water with 3.0–5.0 eV photons. *J. Phys. Chem.* **100**, 17940-17949 (1996).
35. Marsalek O., Elles C. G., Pieniazek P. A., Pluhařová E., VandeVondele J., *et al.* Chasing charge localization and chemical reactivity following photoionization in liquid water. *J. Chem. Phys.* **135**, 224510 (2011).
36. Herbert J. M. Structure of the aqueous electron. *Phys. Chem. Chem. Phys.* **21**, 20538-20565 (2019).
37. Loh Z.-H., Doumy G., Arnold C., Kjellsson L., Southworth S. H., *et al.* Observation of the fastest chemical processes in the radiolysis of water. *Science* **367**, 179-182 (2020).
38. Ambrosio F., Pasquarello A. Reactivity and energy level of a localized hole in liquid water. *Phys. Chem. Chem. Phys.* **20**, 30281-30289 (2018).
39. Ma J., Schmidhammer U., Pernot P., Mostafavi M. Reactivity of the strongest oxidizing species in aqueous solutions: The short-lived radical cation H_2O^+ . *J. Phys. Chem. Lett.* **5**, 258-261 (2014).
40. Schwartz B. J., Rossky P. J. Aqueous solvation dynamics with a quantum mechanical solute: Computer simulation studies of the photoexcited hydrated electron. *J. Chem. Phys.* **101**, 6902-6916 (1994).
41. Larsen R. E., Glover W. J., Schwartz B. J. Does the hydrated electron occupy a cavity? *Science* **329**, 65-69 (2010).
42. Nishitani J., Yamamoto Y.-i., West C. W., Karashima S., Suzuki T. Binding energy of solvated electrons and retrieval of true UV photoelectron spectra of liquids. *Sci. Adv.* **5**, eaaw6896 (2019).
43. Luckhaus D., Yamamoto Y.-i., Suzuki T., Signorell R. Genuine binding energy of the hydrated electron. *Sci. Adv.* **3**, e1603224 (2017).
44. Shreve A. T., Yen T. A., Neumark D. M. Photoelectron spectroscopy of hydrated electrons. *Chem. Phys. Lett.* **493**, 216-219 (2010).
45. Siefermann K. R., Liu Y., Lugovoy E., Link O., Faubel M., *et al.* Binding energies, lifetimes and implications of bulk and interface solvated electrons in water. *Nat. Chem.* **2**, 274-279 (2010).
46. Tang Y., Shen H., Sekiguchi K., Kurahashi N., Mizuno T., *et al.* Direct measurement of vertical binding energy of a hydrated electron. *Phys. Chem. Chem. Phys.* **12**, 3653-3655 (2010).
47. Yamamoto Y.-i., Karashima S., Adachi S., Suzuki T. Wavelength dependence of UV photoemission from solvated electrons in bulk water, methanol, and ethanol. *J. Phys. Chem. A* **120**, 1153-1159 (2016).

REVIEWERS' COMMENTS

Reviewer #1 (Remarks to the Author):

The authors provided a very extensive revision in which they considered my concerns, as well as those of the other two referees, very seriously. They now discuss much better what the novel aspect of this work is. I recommend publication of the paper.

Reviewer #2 (Remarks to the Author):

The authors have addressed in detail all of our comments. I appreciated the expanded discussion section, addressing the alternative hypotheses to explain the observed data. While I would not agree with all the conclusions, it offers the author's perspective and gives the reader a chance to see the present work in the context of previous works.

I still do not quite follow how the authors can exclude the possibility of water molecule excitation to levels below the conduction band. The authors argue that the present signal is not compatible with the previously suggested reaction mechanism at energies below 10 eV. Were there any simulations directly addressing the question of what states are formed upon the string field ionization? Is the experiment sensitive to all of the states formed?

In the above context, I still do not understand fully the NAMD calculations. How was the initial state prepared and how does that compare to the state formed upon the strong field ionization?

Generally, I feel that the authors made an honest effort to improve the manuscript. I do not feel competent to judge how exciting the research is in the view of an extended effort to understand the relaxation dynamics of the hydrated electrons in previous decades. However, the work brings valuable data that will be beneficial for the community.

Reviewer #3 (Remarks to the Author):

The authors have greatly improved their manuscript. I am happy seeing the many suggestions by the reviewers being addressed in sufficient depth. Particularly, the present findings are now well compared with previous works. Certainly, several issues remain still unresolved but from my perspective the new data clearly contributes to a better understanding of the electronic relaxation dynamics of ionized water. I do enjoy reading this latest progress on this topic with a good account on the many related aspects reported in the literature.

There is no need for further changes from my side. However, the authors may want reflect about the following, related to two of my previous comments.

It seems that for some reason the use of CB or CBM (aside from the actual values) is an established quantity even in liquid water. I really do not understand why that should be the case. The meaning of this quantity is not at all clear to me despite the fact that it is used in many works. I assume this is related to the separation distance of 16 solvent shells mentioned on p16.

I appreciate that the authors mention that under their photoionization conditions also photoelectrons will be produced. Indeed, these photoelectrons have no (direct) signature in the reported spectra. But I would argue that the large amount of additional H₂O⁺ (with the electron now in vacuum) must have an effect on the overall electronic relaxation dynamics.

Please, check the following sentence on p17: "These studies yield VDE values of 3.70 ± 0.10 eV⁵², 3.70 ± 0.15 eV³⁹, 3.76 ± 0.05 eV⁵³ and 3.77 ± 0.10 eV⁵³; without correcting for electron scattering, the retrieved VDEs ... " I think the ';' and last ';' must be flipped.

Reviewer #1 (Remarks to the Author):

The authors provided a very extensive revision in which they considered my concerns, as well as those of the other two referees, very seriously. They now discuss much better what the novel aspect of this work is. I recommend publication of the paper.

We are grateful to the reviewer for a careful reading of the revised manuscript and for recommending its publication.

Reviewer #2 (Remarks to the Author):

The authors have addressed in detail all of our comments. I appreciated the expanded discussion section, addressing the alternative hypotheses to explain the observed data. While I would not agree with all the conclusions, it offers the author's perspective and gives the reader a chance to see the present work in the context of previous works.

We are grateful to the reviewer for a careful reading of the revised manuscript and for appreciating the expanded discussion section. In the following, we provide a point-by-point response to the comments raised by the reviewer.

I still do not quite follow how the authors can exclude the possibility of water molecule excitation to levels below the conduction band. The authors argue that the present signal is not compatible with the previously suggested reaction mechanism at energies below 10 eV. Were there any simulations directly addressing the question of what states are formed upon the strong field ionization? Is the experiment sensitive to all of the states formed?

It has been demonstrated in ref. 26 that strong-field ionization of liquid water leads to the vertical injection of an electron into the conduction band. Still, we cannot fully exclude the possibility that the neutral electronically excited states populated by resonance-enhanced strong-field multiphoton ionization might also contribute to the observed ultrafast dynamics, especially when our experiments are insensitive to the transient population of these excited states. However, we note that any neutral electronically excited state that is accessed via multiphoton excitation would be rapidly ionized – on a sub-cycle timescale – in the presence of an intense laser field. We have clarified this on p. 4 of the revised manuscript. **(REV 2.1)** Finally, please note that our ab initio simulations do not address the nature of states that are produced by strong-field ionization. Such analysis requires different theoretical tools and extends beyond the scope of the current communication.

In the above context, I still do not understand fully the NAMD calculations. How was the initial state prepared and how does that compare to the state formed upon the strong field ionization?

The initial state was prepared by populating the region of the conduction band that is located 1 eV above the conduction band minimum of liquid water at equilibrium, mimicking the vertical injection of the electron into the conduction band by the intense laser field. We point the reviewer to second paragraph of the “Theoretical” sub-section of the “Methods” section for this information.

Generally, I feel that the authors made an honest effort to improve the manuscript. I do not feel competent to judge how exciting the research is in the view of an extended effort to understand the relaxation dynamics of the hydrated electrons in previous decades. However, the work brings valuable data that will be beneficial for the community.

We are grateful to the reviewer for acknowledging our efforts to improve the manuscript and for recognizing our data as being valuable to the community.

Reviewer #3 (Remarks to the Author):

The authors have greatly improved their manuscript. I am happy seeing the many suggestions by the reviewers being addressed in sufficient depth. Particularly, the present findings are now well compared with previous works. Certainly, several issues remain still unresolved but from my perspective the new data clearly contributes to a better understanding of the electronic relaxation dynamics of ionized water. I do enjoy reading this latest progress on this topic with a good account on the many related aspects reported in the literature.

There is no need for further changes from my side. However, the authors may want reflect about the following, related to two of my previous comments.

We are grateful to the reviewer for a careful reading of the revised manuscript and for recognizing the contribution of our data to a better understanding of the electronic relaxation dynamics of ionized water.

It seems that for some reason the use of CB or CBM (aside from the actual values) is an established quantity even in liquid water. I really do not understand why that should be the case. The meaning of this quantity is not at all clear to me despite the fact that it is used in many works. I assume this is related to the separation distance of 16 solvent shells mentioned on p16.

We thank the reviewer for this comment. Our use of the term “conduction band” in the manuscript follows the terminology from the literature, stemming from condensed matter physics of periodic systems. It relates to the spatially delocalized nature of the unoccupied states. Although unoccupied states of bulk water are not fully delocalized and are not periodic, they are delocalized over many water molecules, allowing one to use the “conduction band” terminology in a loose sense.

I appreciate that the authors mention that under their photoionization conditions also photoelectrons will be produced. Indeed, these photoelectrons have no (direct) signature in the reported spectra. But I would argue that the large amount of additional H₂O⁺ (with the electron now in vacuum) must have an effect on the overall electronic relaxation dynamics.

We thank the reviewer for this comment. Since vertical ionization requires ~1-eV higher energy input than vertical injection of an electron into the conduction band, the relative contributions of the two channels can be changed by varying the ionizing pump power. As such, a future study of the influence of the ionizing pump power on the observed ultrafast dynamics will potentially allow us to elucidate the effect of excess H₂O⁺ on the electronic relaxation dynamics.

Please, check the following sentence on p17: “These studies yield VDE values of 3.70 ± 0.10 eV⁵², 3.70 ± 0.15 eV³⁹, 3.76 ± 0.05 eV⁵³ and 3.77 ± 0.10 eV⁵³; without correcting for electron scattering, the retrieved VDEs ... “ I think the ‘;’ and last ‘,’ must be flipped.

The semi-colon separates the list of vertical detachment energy (VDE) values from the statement about retrieved VDEs being too low in the absence of corrections for electron scattering.